# Cognitive and Non-Cognitive Predictors of Response to Cognitive Stimulation Interventions in Dementia: A Systematic Review Aiming for Personalization [note 1]

**DOI:** 10.3390/bs15081069

**Published:** 2025-08-06

**Authors:** Ludovica Forte, Giulia Despini, Martina Quartarone, Lara Calabrese, Marco Brigiano, Sara Trolese, Alice Annini, Ilaria Chirico, Giovanni Ottoboni, Maria Casagrande, Rabih Chattat

**Affiliations:** Department of Psychology, University of Bologna, 40126 Bologna, Italy; dott.giuliadespini@gmail.com (G.D.); martina.quartarone2@unibo.it (M.Q.); lara.calabrese3@unibo.it (L.C.); marco.brigiano2@unibo.it (M.B.); sara.trolese@unibo.it (S.T.); alice.annini2@unibo.it (A.A.); ilaria.chirico2@unibo.it (I.C.); giovanni.ottoboni@unibo.it (G.O.); maria.casagrande@uniroma1.it (M.C.)

**Keywords:** dementia, cognitive intervention, cognitive stimulation therapy, cognitive training, randomized controlled trial

## Abstract

Despite the extensive evidence supporting the effectiveness of cognitive stimulation, differences in results may be due to the influence of cognitive and non-cognitive aspects in people with dementia. The aim of this systematic review is to identify the most reliable variables in forecasting the effectiveness of cognitive stimulation in people with mild to moderate dementia. According to PRISMA guidelines, the research was conducted using five databases (PubMed, Scopus, Cochrane, Web of Science, APA PsycInfo), considering randomized controlled trials. A total of six studies were included. Different aspects moderating the gain resulting from cognitive intervention were collected and assessed in terms of demographic, cognitive, emotional, social, and quality of life parameters. People with dementia benefit more from cognitive intervention if they are female, if they have a low formal education level, a low baseline level of cognitive function, and lower depressive symptoms, and if caregivers actively participate in sessions. Quality of life, if low at baseline, also seems to improve following CST intervention. A deeper understanding of the cognitive and non-cognitive aspects ensuring improvement after cognitive stimulation may guide future research to develop more personalized interventions.

## 1. Introduction

Life expectancy is increasing due to advancements in healthcare, improved living conditions, public health policies, and better disease prevention: according to the World Health Organization (WHO), the global population over the age of 60 is projected to reach 2 billion by 2050. Consequently, the continuous rise in life expectancy has contributed to a growing prevalence of age-related diseases. Among these, dementia represents one of the most significant challenges in terms of diagnosis, treatment, and care ([26]), making it a major concern for current healthcare systems. Data from the WHO Global Action Plan 2017–2025 (Global action plan on the public health response to dementia 2017–2025; [75]) indicate that, dementia will affect 75 million people worldwide by 2030 and 132 million by 2050, with approximately 10 million new cases per year (1 every 3 s). 

The available drugs for the treatment of cognitive symptoms in people with dementia (particularly Alzheimer’s disease and Lewy body dementia) are cholinesterase inhibitors (donepezil, rivastigmine, galantamine) and memantine. These are associated with a reduction in symptom severity and with a slower decline in cognition and activities of daily living, as well as decreased mortality in individuals with both mild to moderate and severe dementia ([35]). The latest significant advance in disease-modifying therapies for dementia—particularly Alzheimer’s disease—is the conditional approval of donanemab and lecanemab, two monoclonal antibodies that reduce amyloid-β plaque levels. However, evidence supporting meaningful clinical benefit—such as significantly slowed cognitive decline—remains limited, and the association between amyloid reduction and improved clinical outcomes is weak ([35]). However, these pharmacological discoveries are not considered cost-effective, as they offer only modest benefits, only affect a small subset of patients, and may cause serious adverse effects in some cases ([8]). Therefore, research has focused on the development and application of non-pharmacological treatments, in particularly psychosocial interventions, increasingly considered a core component of dementia care ([29]; [36]; [60]). Psychosocial approaches represent a valuable non-pharmacological alternative for managing cognitive decline, behavioral symptoms, and overall quality of life in people with dementia ([59]). Among the various psychosocial interventions—such as behavioral, environmental, and cognitive approaches—cognitive stimulation (CS) has shown the most consistent benefits in people with mild to moderate dementia ([73]; [40]). The 2024 update of the Lancet Commission ([35]) also includes cognitive interventions among the recommended approaches for managing cognitive symptoms in dementia intervention and care. 

Within the broad category of CS programs, Cognitive Stimulation Therapy (CST) stands out as a structured and manualized group-based intervention specifically developed for this population ([62]). CST is supported by robust evidence demonstrating positive effects not only on global cognition (intended here as overall cognitive functioning measured through standardized cognitive screening tools that provide a general overview of cognitive abilities without focusing on specific domains), but also on specific cognitive domains such as memory, attention, language, orientation, neuropsychiatric symptoms, daily functioning, and quality of life ([2]; [30]; [19]; [76]). For these reasons, CST is currently the only psychosocial intervention formally recommended to improve cognition in people with dementia by both the UK’s National Institute for Health and Care Excellence ([44]) and the Italian national guidelines ([27]). It is also widely considered a cost-effective gold-standard intervention in non-pharmacological dementia care ([31]; [40]). Despite the growing body of literature supporting the effectiveness of cognitive stimulation interventions, findings remain partially inconsistent across studies. For example, [33] ([33]) reported non-significant effects on cognitive function, while [10] ([10]) found ambiguous results regarding activities of daily living, and [72] ([72]) highlighted mixed evidence regarding depressive symptoms and quality of life. One possible explanation for this variability may lie in the individual characteristics of people with dementia, including both cognitive factors (e.g., baseline cognitive performance) and non-cognitive ones (e.g., emotional status, social engagement, demographic variables), which may influence the outcomes of such interventions. This observation highlights the importance of further investigating which individual-level predictors may moderate or mediate the effectiveness of cognitive stimulation and lead to the development of more personalized interventions. In recent years, research has increasingly focused on the personalization of cognitive interventions. [38] ([38]) defined the personalization of tailored activities “as the extent to which non-pharmacological interventions are tailored, individualized and personalized for people with dementia,” in a person-centered approach. According to these authors, personalization depends on the assessment of individual’s cognitive, functional, and affective characteristics. Evidence from dementia-specific RCTs and reviews supports the efficacy of personally tailored activities, which are adapted based on individual interests, functional abilities, cognitive functioning, or emotional needs. For example, tailoring may involve adapting tasks to one’s level of cognitive functioning, personal interests, or emotional needs. Personalized activities are associated with greater engagement and adherence, as they make the experience more meaningful and empowering for people with dementia, promoting participation and improved outcomes ([39]; [68]; [7]). A Cochrane review of community-dwelling individuals with dementia found moderate evidence that such personalized interventions reduce challenging behavior and slightly improve quality of life ([54]; [18]). Additionally, a case report by [50] ([50]) demonstrated that a culturally and psychologically personalized cognitive stimulation program enhanced engagement, symptom acceptance, and mild cognitive benefit. Furthermore, the SADEM trial in adults with mild dementia showed that a long-term (12-month) multicomponent cognitive intervention produced improvements in cognition and daily living activities ([25]). These findings underscore the importance of personalization, even in traditional non-pharmacological interventions, to enhance relevance and effectiveness for individuals with dementia.

The goal of this review was to study the influence of cognitive and non-cognitive aspects to better understand which variables are associated with more favorable outcomes and which individuals are more likely to benefit from cognitive stimulation interventions. 

## 2. Materials and Methods

This study was registered in the International Prospective Register of Systematic Review PROSPERO (CRD42024520857) on 5 March 2024. A systematic review was conducted under the Preferred Reporting Items for Systematic Reviews and Meta-Analysis guidelines (Prisma) ([53]).

### 2.1. Search Strategy

A systematic literature search was performed to identify relevant studies, using the following electronic databases, between February and March 2024, with an update to March 2025: PubMed, Web of Science, PsycINFO, Scopus, and Cochrane. The search strategy used the syntax reported in the following box: (“cognitive stimulation” OR “cognitive stimulation therapy” OR “cognitive training” OR “cognitive rehabilitation” OR “cognitive intervention”) AND (“dementia” OR “Alzheimer” OR “lewy body dementia” OR “frontotemporal dementia” OR “vascular dementia”).



### 2.2. Inclusion Criteria

Specific criteria were established according to the PICOs principles, a framework used to formulate well-defined research questions, facilitating the search for relevant evidence and improving the quality of research. PICOs stands for Participants (which refers to the specific patient group or population being studied), Interventions (which describes the intervention being investigated), Comparison/Control (which identifies the comparison group or control group, which may be another intervention, a placebo, or no intervention at all), Outcomes (specifies the outcome being measured or assessed), and Study Design (refers to the type of studies being considered). The inclusion criteria were as follow [Table 1]:(a)studies that included participants older than 65 years of age and with a diagnosis of mild to moderate dementia. From a clinical and epidemiological standpoint, dementia occurring before the age of 65 is often classified as young-onset dementia (YOD) ([57]), also known as early-onset dementia (EOD) ([28]), which may involve different etiologies, care needs, and psychosocial implications compared to late-onset dementia ([21]; [67]). So, including younger individuals would therefore have introduced significant heterogeneity and potentially confounded our analysis. We also restricted inclusion to individuals with mild to moderate dementia, as cognitive stimulation interventions are specifically recommended for this subgroup. Evidence from both clinical trials and international guidelines indicates that cognitive stimulation interventions are most effective—and most appropriate—for people with mild to moderate levels of cognitive impairment ([62]; [44]). In people with severe dementia, cognitive stimulation interventions may be less feasible or have diminished efficacy due to greater functional limitations and lower cognitive reserve.(b)studies that included isolated cognitive stimulation treatment, according to Clare and Woods’ definition of cognitive stimulation as “engagement in a range of group activities and discussions aimed at general enhancement of cognitive and social functioning,” rather than interventions targeting a specific cognitive function. This type of multi-domain, non-specific stimulation is distinct from cognitive training (which targets specific domains such as memory or attention), cognitive rehabilitation (which is goal-oriented and individualized), and other psychosocial interventions ([14]; [62]; [73]). This category includes both standardized Cognitive Stimulation Therapy (CST) protocols as well as other cognitive stimulation interventions sharing similar principles but not strictly following manualized CST.(c)studies that included a passive control group that received standard care (treatment as usual) or no active treatment. This choice was made to ensure that any observed effects could be more confidently attributed to the cognitive stimulation intervention, minimizing potential confounding effects introduced by other simultaneous activities.(d)studies which evaluated multiple outcome domains—including functional, cognitive, psychological, and affective outcomes—assessed both before and after cognitive stimulation intervention and that underlined the influence of cognitive and non-cognitive aspects of people with dementia on the gains resulting from cognitive stimulation intervention. We specifically included studies that evaluated the effects of cognitive stimulation on multiple cognitive domains rather than on a single cognitive function, because dementia typically affects a range of cognitive abilities simultaneously. Measuring improvement across multiple domains allows for a more comprehensive and ecologically valid assessment of the intervention’s effectiveness. This approach aligns with existing literature emphasizing the importance of multi-domain cognitive assessments in dementia research to detect meaningful and generalizable changes ([14]; [77]). Therefore, our inclusion criteria aimed to capture studies that evaluate cognitive outcomes more comprehensively to inform the development of personalized and effective cognitive stimulation interventions. Furthermore, a key inclusion criterion was that studies had to explore the influence of individual cognitive and/or non-cognitive characteristics (e.g., baseline cognitive level, mood, education, age) on the outcomes of the intervention. This focus reflects the increasing recognition in dementia care and research of person-centered approaches, which emphasize the importance of identifying which individuals are more likely to benefit from specific interventions ([13]; [77]).(e)studies whose design was that of a randomized controlled trial (RCT) to select studies with demonstrated evidence of efficacy with higher standards. RCTs are widely considered the gold standard for evaluating the efficacy of interventions and for minimizing selection bias and confounding ([42]). This choice was made to ensure that the evidence reviewed was based on robust and methodologically sound designs capable of supporting causal inferences.

### 2.3. Exclusion Criteria

The exclusion criteria were:(a)studies focusing on an adult population younger than 65 years, or patients without a diagnosis of dementia, or those presenting with other medical or psychiatric conditions such as major psychiatric disorders, stroke, or traumatic brain injury. This last decision was grounded in standard diagnostic criteria for dementia. DSM-5 and ICD-10 require that cognitive decline not be better explained by other neurological, psychiatric, or systemic medical disorders ([5]; [70]).(b)studies that included isolated cognitive stimulation interventions focused on a single cognitive function (e.g., memory-only tasks), multifactorial intervention (e.g., combining physical activity, diet, and cognitive tasks without isolating the cognitive stimulation component), other psychosocial interventions (e.g., reminiscence therapy or Reality Orientation Therapy (ROT)) that do not involve cognitive stimulation, or combined cognitive stimulation interventions with pharmacological treatments, as this would have made it difficult to isolate the effect of the cognitive component. The decision to exclude studies focusing on cognitive stimulation targeting a single cognitive domain is supported by existing literature emphasizing the superior efficacy of multi-domain cognitive stimulation interventions. [14] ([14]) clarify that cognitive stimulation is characterized by engaging multiple cognitive domains, distinguishing it from domain-specific cognitive training or rehabilitation. Systematic reviews, including those by [73] ([73]) and [6] ([6]), provide evidence that multi-domain cognitive stimulation leads to broader improvements in cognitive functioning and daily living activities compared to interventions focusing on a single cognitive domain. Furthermore, clinical guidelines such as those from [44] ([44]) recommend multi-domain approaches as standard practice for cognitive interventions in dementia care, reinforcing the rationale behind our exclusion criteria.(c)studies with an active control group, such as those engaging participants in alternative cognitive, social, or behavioral activities (e.g., recreational groups, psychoeducation, or other non-specific engagement strategies). Active controls, while useful in some contexts, may reduce the ability to isolate the unique contribution of cognitive stimulation, particularly in a systematic review aiming to explore moderators of response.(d)studies that did not include cognitive and non-cognitive outcomes or that focused solely on improvement of a single cognitive function or did not explore the relationship between individual characteristics and intervention outcomes (e.g., studies reporting only overall group-level effects, without analysis of influencing factors).(e)studies without a control group or with only pre- and post-treatment comparison within a single group.(f)furthermore, protocol studies, abstracts, or posters from congresses and studies with no full text available were excluded.

### 2.4. Study Selection and Data Extraction

Three reviewers (L.F., G.D., M.Q.) performed the database search. All the studies were entered into RAYYAN, a systematic literature review tool that helps researchers in the screening of articles. The first exclusion of irrelevant studies was made by analyzing the titles and abstracts of the articles (L.F., G.D., M.Q., L.C., M.B., S.T., A.A., I.C.) and subsequently reading the full text (L.F., G.D., M.Q., L.C., M.B., S.T., I.C.). A single reviewer (L.F.) conducted the extraction of key data from those final studies, considering author, publication year, country, study design, participants’ demographic data, the experimental and control group included in the intervention, the tools used to evaluate outcomes, and the main results. However, all extracted information was subsequentially reviewed by all co-authors (G.D., M.Q., L.C., M.B., S.T., A.A., I.C.), who were thoroughly familiar with the included studies. This collaborative process allowed for cross-checking of data and minimization of potential errors or biases. Any discrepancies or inconsistent decisions were resolved by discussion and by consulting a supervisor (R.C.).

### 2.5. Study Risk of Bias Assessment

The risk of bias was evaluated for each study using the Mixed Method Appraisal Risk (MMAT) Version 2018 risk-of-bias tool ([24]). This tool includes seven domains which allow appraisal of the quality of the studies in terms of the clarity of the research questions, the randomization process, group comparison at baseline, incomplete outcome data, blinding of the outcome assessors, and participants’ adhesion to the assigned intervention. Domains that were fulfilled were marked with a “Yes,” while those that were not met were marked with a “No.” “Cannot tell” was used when the study did not provide sufficient information. The risk of bias assessment was conducted by one author (L.F.) and reviewed by a second author (M.Q.); possible differences in the assessments would have been checked by a third reviewer (R.C.).

## 3. Results

### 3.1. Study Selection

A total of 11,785 records were identified through database searching. All records were entered into RAYYAN, and duplicates (n = 4371) were removed, leaving 7414 studies. After excluding irrelevant studies by analyzing titles and abstracts (n = 7162), the full texts assessed for eligibility were 252. After full text review, 246 articles did not to meet the PICOs method criteria, as follows: the study population was not the population of interest for this review (n = 47); the study did not implement a cognitive stimulation intervention (n = 40); the outcome of the study was not of interest for this review (n = 60); the study had an inappropriate study design (n = 26); and the publication type was not suitable (n = 73). Ultimately, six studies met all inclusion criteria and were included in the final analysis: Cove et al., 2014; Kwok et al., 2013; Middelstadt et al., 2016; Neely et al., 2009; Paddick et al., 2017; and Quayhagen et al., 1995. The complete study selection process is summarized in the PRISMA flow diagram [Figure 1].

### 3.2. Risk of Bias Analysis

The risk of bias was assessed using the Mixed Method Appraisal Risk (MMAT) Version 2018 risk-of-bias tool. All six included studies met the criteria outlined in the first two screening questions (S1 and S2). All were quantitative randomized controlled trials. All studies reported complete outcome data (question 2.3), and participants generally adhered to the assigned intervention (question 2.5). However, three studies ([32]; [45]; [56]) did not provide detailed information on participant allocation (question 2.1), making it unclear whether randomization was properly performed. In contrast, the remaining three studies used computer-generated random numbers or a random number list for allocation. With the exception of one study ([32]), where the authors acknowledged baseline differences in age, education level, and CSSA status between the intervention and control groups, the groups in all the other studies were well balanced at baseline (question 2.2). Blinding of the outcome assessors (question 2.4) was not clearly reported in two studies ([45]; [56]). Overall, all included studies were judged to have a low risk of bias. The quality appraisal domains for each study are summarized in Table 2.

### 3.3. Study Descriptions

All six studies in this systematic review were randomized controlled trials (RCTs), conducted in different areas of the world: Germany ([41]), the USA ([56]), the United Kingdom ([15]), sub-Saharan Africa ([52]), Sweden ([45]), and Hong Kong ([32]). In total, these studies involved 457 participants, with sample sizes ranging from 30 ([45]) to 176 ([32]). The majority of participants were women (313 women vs. 144 men). The mean age of participants varied between 73.6 years ([56]) and 86.37 years ([41]). All studies included individuals with a clinical diagnosis of mild to moderate dementia, although diagnostic criteria differed. Three studies ([45]; [15]; [52]) used the Diagnostic and Statistical Manual of Mental Disorders, Fourth Edition (DSM-IV) criteria for dementia of any type. Two studies ([32]; [56]) applied cut-off scores from cognitive screening tests: [32] ([32]) included participants scoring 23 or higher on the Mini-Mental State Examination (MMSE), while [56] ([56]) required a score of 90 or above on the Mattis Dementia Rating Scale. [41] ([41]) diagnosed dementia according to ICD-10 criteria. Interventions took place in various settings serving older adults, including long-term care facilities ([41]), community-dwelling services ([56]; [15]), an Alzheimer’s village ([52]), and district elderly community centers ([32]). One study delivered the intervention at participants’ homes ([45]). The content of the cognitive stimulation interventions varied but generally included active cognitive exercises targeting memory, language, problem solving, and executive function, as well as sensory stimulation and social interaction activities. In two studies ([52]; [32]), the CST content was culturally and educationally adapted to fit the specific populations and available resources. The frequency of the cognitive stimulation intervention ranged from once a week ([15]; [45]; [32]) to daily sessions ([56]). Session durations were 45 min ([15]) or 60 min ([41]; [56]; [45]; [32]), and total intervention lengths varied from 1 week ([56]) to 14 weeks ([15]). A summary of the study characteristics is presented in Table 3. 

### 3.4. Cognitive and Non-Cognitive Factors Associated with Greater Benefit from Cognitive Stimulation

From the six included studies, factors potentially influencing the benefits gained from cognitive stimulation intervention were extracted a posteriori then assessed in terms of demographic, cognitive, emotional, and social factors and quality of life [Table 4].

*Socio-demographic factors.* [56] ([56]), involving 78 people with dementia, identified gender as a significant factor influencing intervention outcomes: men showed a decline in general memory performance over time, whereas women maintained or improved their performance. Two studies, encompassing a total of 210 participants, highlighted the impact of education level ([52]; [32]). [52] ([52]) reported greater cognitive improvement immediately post-intervention, measured by the ADAS-Cog, among individuals with no formal education. Similarly, [32] ([32]) found that participants with lower education levels or illiteracy exhibited greater cognitive gains, as assessed by the Chinese version of the Mattis Dementia Rating Scale (CDRS), following the Active Mind program. These findings suggest that gender and education level are key demographic factors associated with better responses to cognitive stimulation.

*Cognitive factors.* Two studies including 139 people with dementia identified baseline cognitive functioning as a predictor of intervention benefit ([41]; [15]). [41] ([41]) found a significant three-way interaction effect (ADAS-Cog baseline x Depression x Group), indicating that participants with lower baseline cognitive scores and fewer depressive symptoms were more likely to show cognitive improvements according to the ADAS-Cog from pre-test to follow-up. Conversely, [15] ([15]) observed no cognitive gains, attributing this to their sample’s higher baseline cognitive functioning, as assessed by the MMSE and the ADAS-Cog, suggesting that cognitive stimulation may be less effective for individuals with relatively preserved cognition.

*Emotional factors.* Depression was identified as a moderator of cognitive and quality-of-life outcomes in the study by [41] ([41]). A significant three-way interaction effect (ADAS-Cog baseline x Depression x Group) showed that participants with lower baseline scores for cognitive performance and fewer depressive symptoms had a higher probability of showing cognitive improvements, as assessed by the ADAS-Cog, from pre-test to follow-up. Conversely, participants with higher depression levels and low cognitive baseline had the lowest likelihood of cognitive gains. Another significant three-way interaction effect (QoL-AD Proxy Baseline x Depression x Group) showed that participants in the experimental group with low quality of life and low depressive symptoms at baseline had a higher probability of improving their QoL from pretest to follow up. In other words, those patients who had a low baseline level of cognition and QoL benefitted most in terms of the corresponding variables at follow-up if they had low depression scores at the beginning of the intervention.

*Social factors.* [45] ([45]), involving 30 dyads of people with dementia and their caregivers, demonstrated that the presence of a caregiver during cognitive stimulation sessions enhanced intervention benefits. Participants in the collaborative program, which involved joint sessions with caregivers, showed greater cognitive improvements compared to those in the individual training or control groups. People with dementia in the individual group received the same amount of cognitive training as the participants with dementia in the collaborative program but did not improve their performances as a function of the training. This suggests that the social and interactive nature of the collaborative approach may contribute to better cognitive outcomes: working with a caregiver was beneficial for cognition.

*Quality of life.* [41] ([41]) found a significant three-way interaction effect (QoL-AD Proxy Baseline x Depression x Group), showing that participants in the experimental group with low QoL (externally assessed) and low depressive symptoms at baseline had a higher probability of improving their QoL (externally rated) from pre-test to follow up. So, a low baseline level of externally rated QoL predicted improvements in externally rated QoL assessed after cognitive training, and this was also moderated by depression.

## 4. Discussion

This systematic review investigated cognitive and non-cognitive aspects of people with dementia that can represent the most reliable variables in forecasting the effectiveness of CST intervention. Only six articles were considered suitable for inclusion in this study. The included articles describe different aspects that most influence participants’ cognitive gain after cognitive stimulation. 

Among the demographic variables, the main findings are that people with dementia who are women ([56]) and have a low education level ([52]; [32]) are the ones who can gain more benefits from cognitive training. [2] ([2]) also associated the female gender with greater improvements in cognition, pointing out that older age and being female were associated with increased cognitive benefits from CST. Regarding level of education, several studies found more benefits from cognitive stimulation in people with dementia with a low education level ([47]; [30]; [11]). A possible explanation is that, in people with no formal education, it is easier to see improvements. Another reason is possible if the theoretical framework of the cognitive reserve paradigm is considered: education and other activities are proposed to increase cognitive reserve and the ability to cope with dementia’s symptoms. According to [63] ([63], [64], [65]), there is a point after which cognitive reserve can no longer withstand the pathology, and the progression of the disease is dramatically faster. A high cognitive reserve delays the occurrence of symptoms; however, when the severity of dementia is high, people with dementia are so impaired that they cannot benefit from cognitive stimulation interventions. [43] ([43]) investigated the influence that cognitive reserve may have on modulating the efficacy of a cognitive training and found improvements in subjects with a lower level of cognitive reserve, while patients with a higher cognitive reserve do not seem to benefit from the intervention.

Another finding is that a low baseline level of cognitive function predicted improvements in cognition ([41]; [15]). The cognitive and global functioning level of people with dementia may predispose them to differences in the effectiveness of treatment: different studies states that people with dementia with higher cognitive functioning may derive less benefit from cognitive stimulation interventions ([15]; [20]). As a possible interpretation, [69] ([69]) suggested that persons with lower cognitive abilities at baseline have greater scope for improvement. 

Moreover, depression moderates the effect of cognitive level and QoL level on the effectiveness of CST intervention. In [41] ([41]), people with dementia with low depressive symptoms, lower cognitive baseline performance, and low baseline QoL level showed the highest benefits. The explanation from the authors was that persons without depressive symptoms are more likely to be able to process motivating intervention content with possible positive effects. Depression is named as one of the BPSD (behavioral and psychiatric symptoms of dementia) group symptoms and is associated with reduced motivation and impaired self-confidence ([17]): it is likely that these factors make it difficult to benefit from cognitive stimulation intervention. 

Previous literature already examined the influence of social relationships on the progression of dementia. More generally, being married, exchanging support with family members, maintaining contact with friends, and participating in community groups were all related to a lower likelihood of developing dementia ([58]). It is also known that conducting the intervention in groups can encourage social interaction ([9]; [47]) and increase cognitive gain ([62]) and quality-of-life benefits for people with dementia ([37]), unlike individually conducted interventions ([34]). According to [4] ([4]), the improvements observed in people with different degrees of severity following cognitive stimulation intervention were mainly due to social involvement. The results from the interviews of [49] ([49]) support the advantages derived from a group-based approach to CST, where people with dementia seem to experience sharing a common difficulty related to the disease as helpful ([61]). So, these results suggest that dementia may potentially be prevented by enhancing these social relationships and the social context. Analogous results are observed when considering other type of relationships, such as a closer relationship or a marital relationship. It is known that there is a significant association between the presence of a spouse and incident dementia ([58]). In a closer relationship, people with dementia declined more slowly in cognition and functional ability; this was more evident when caregivers were the spouses of the people with dementia ([46]). [45] ([45]) found that involving caregivers in cognitive training sessions has a more positive effect on the cognitive performance of spouses with dementia, and this benefit could partly be derived from the more social interactive context of a program that includes caregivers rather from the program’s content itself. Some research suggests that interventions that focus on collaborative aspects of care dyads show positive benefits for both caregivers and people with dementia, resulting in additional benefits for people with dementia if caregivers apply its principles and use cognitive stimulation activities between sessions ([48]; [55]). [51] ([51]) reported that caregivers’ confidence in their ability to adopt a therapeutic role may also affect the success of the intervention. 

Lastly, baseline level of quality of life appears to be both a moderator and a predictor of the benefit gained from cognitive stimulation. In [41] ([41]), people with dementia who had low baseline levels of cognition and QoL benefitted most in terms of the corresponding variables, moderated by a low depression score at the beginning. [74] ([74]) also showed that improvements in quality of life were associated with low quality of life at baseline, reduced depression, and increased cognitive function. Several studies have shown that the progression of dementia negatively affects quality of life, so encouraging dementia patients to participate in challenging activities such as those involved in cognitive stimulation can improve their quality of life ([23]). According to [4] ([4]), cultural aspects may conceal the effects of cognitive stimulation interventions on the perception of quality of life, and consequently the same negative stereotypes about dementia could have a detrimental impact on cognitive function ([22]). In a study by [78] ([78]), social workers who conducted the cognitive stimulation intervention adopted a destigmatizing approach by promoting acceptance of illness, valuing people with dementia, and encouraging participants to express their opinions and views. It seems that this destigmatizing approach contributes to the effectiveness of cognitive stimulation interventions. 

So far, we have only discussed the cognitive and non-cognitive characteristics of the people with dementia. [15] ([15]) found another aspect that could modulate the effectiveness of cognitive interventions and showed that it could be assessed as a characteristic of the intervention context. The authors reported that a cognitive intervention delivered once a week may not offer the necessary “dose” necessary to combat cognitive decline: it is too brief to have made an improvement. In effect, the standardized program of [62] ([62]) includes cognitive stimulation therapy two times a week. 

Despite some heterogeneity among the included studies in terms of CST delivery, outcome measures, cultural context, and intervention duration and dose, these differences do not appear to undermine the consistency of the findings. Regarding the intervention setting, previous literature has shown that the effectiveness of CST is not significantly influenced by whether it is delivered in community-based environments, in care homes, or at home. A Cochrane review by [73] ([73]) found no significant subgroup differences in outcomes based on setting, supporting the generalizability of CST interventions across diverse environments. With respect to cultural context, although the studies included in this review were conducted in different countries, evidence from large-scale systematic reviews and culturally adapted CST programs suggests that the intervention maintains its efficacy across diverse populations. For example, [73] ([73]) analyzed 33 RCTs across 17 countries and five continents, reporting consistent cognitive and quality-of-life improvements. Furthermore, [1] ([1]) proposed the Formative Method for Adapting Psychotherapy (FMAP) to guide the cultural adaptation of CST in low- and middle-income countries, ensuring that the intervention remains structurally consistent. Pilot studies from Hong Kong ([71]), Portugal ([3]), and Māori communities in New Zealand ([16]) further support the cross-cultural applicability of CST, with adaptations enhancing local acceptability without compromising core elements. Although different cognitive and non-cognitive outcome measures were used across the included studies, these tools were largely aimed at assessing the same domains—particularly global cognitive function (e.g., MMSE, ADAS-Cog), specific cognitive abilities, and quality of life. This suggests that, despite the diversity of instruments, the studies maintained conceptual consistency in their outcome assessment. Lastly, differences in CST duration and dose reflect real-world implementation rather than methodological inconsistency. While the original protocol suggests 14 sessions over 7 weeks ([62]), recent meta-analyses ([12]; [66]) have confirmed that CST maintains its efficacy even when session number or frequency varies moderately. In fact, most of the included studies in our review provided interventions lasting approximately 60 min per session, and the number of sessions generally ranged from 8 to 14. This suggests that CST can be adapted flexibly in duration and format without a significant loss of efficacy. 

We hope that future research will consider these characteristics to improve personalized CST interventions: the existing literature demonstrates that personalization of tailored activities maintains adherence to the intervention and, consequently, leads to improved outcomes. 

Unfortunately, there is a lack of available studies on this topic, so we could not investigate the influence of other aspects on CST effectiveness, such as personality traits, other BPSDs besides depression, and type or severity of dementia.

## 5. Limitations and Strengths

Although this systematic review highlighted important insights regarding the influence of cognitive and non-cognitive factors on the effectiveness of cognitive stimulation, it also has some limitations. First, the use of stringent inclusion criteria ensured high methodological quality but also considerably limited the number of eligible studies, thus reducing the generalizability of the findings. For instance, only randomized controlled trials were included to maintain methodological rigor, which resulted in the exclusion of many potentially relevant studies that did not meet this design criterion. Additionally, some articles were excluded because their sample populations were slightly younger than the specified age range or because they were study protocols rather than completed trials. This focused scope, while limiting the number of included studies, was intentional and aligned with the specific aim of identifying predictive factors, rather than evaluating general efficacy or maximizing generalizability.

Despite these limitations, the six included articles demonstrated overall good methodological quality, except for some weakness in the randomization process. A key strength of this review is its rigorous methodological approach, which ensures the reliability of the findings presented. Moreover, by concentrating specifically on cognitive and non-cognitive predictors of deriving benefit from cognitive stimulation, this review offers a focused and novel contribution to the literature on psychosocial interventions in dementia. Finally, the review highlights the need for further research exploring a wider range of factors that may influence the effectiveness of cognitive stimulation interventions in people living with dementia.

## 6. Conclusions

Cognitive stimulation is a non-pharmacological treatment that is considered a gold standard in dementia care, with consistent evidence of positive outcomes for people with mild to moderate dementia. The results of this systematic review provide insight into which cognitive and non-cognitive aspects of people with dementia are associated with greater benefit from a cognitive stimulation intervention. Collectively, these RCT observations suggest that older patients, female gender, individuals with low formal education level, a low baseline level of cognitive function, and lower depressive symptoms may be associated with greater improvements from cognitive stimulation intervention. Moreover, people with dementia benefit more if caregivers actively participate in cognitive stimulation sessions. Low quality of life level at baseline also seems to improve, together with cognition, following cognitive stimulation if there are fewer depressive symptoms. 

We suggest that these aspects be considered to improve personalization of cognitive stimulation programs, tailoring the interventions to individual characteristics in order to maximize their effectiveness.

## Figures and Tables

**Figure 1 behavsci-15-01069-f001:**
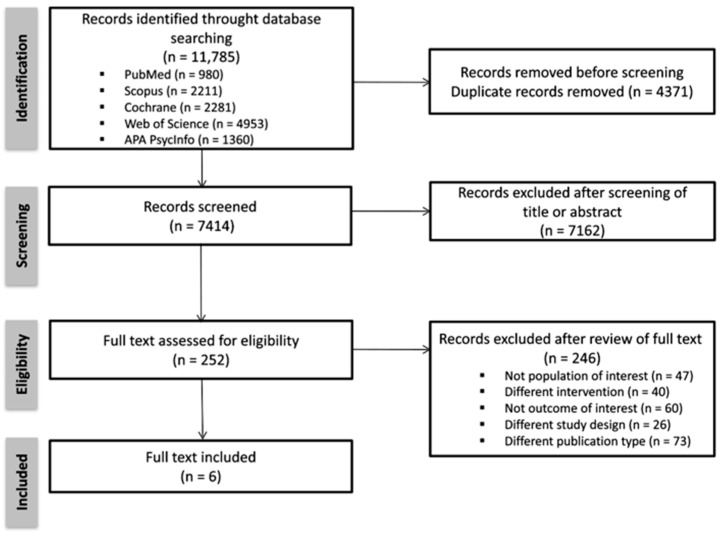
PRISMA flow diagram.

**Table 1 behavsci-15-01069-t001:** PICOs criteria.

Parameters	
Participants	Adults aged 65 years old or more and with a diagnosis of mild to moderate dementia.
Interventions	Cognitive Stimulation (CS) interventions (including manualized CST protocols and other cognitive stimulation programs).
Control	Treatment as usual (standard care) or no active treatment.
Outcomes	Functional, cognitive, psychological, and affective outcomes evaluated before and after the cognitive stimulation intervention. Influence of cognitive and non-cognitive aspects of people with dementia on the gains resulting from cognitive stimulation intervention.
Study design	Randomized controlled trials (RCTs).

**Table 2 behavsci-15-01069-t002:** Risk of bias assessment using MMAT 2018.

Category of Study Design	Methodological Quality Criteria	[15] ([15]) Quantitative RCT	[32] ([32]) Quantitative RCT	[41] ([41]) Quantitative RCT	[45] ([45]) Quantitative RCT	[52] ([52]) Quantitative RCT	[56] ([56]) Quantitative RCT
1. Screening questions	S1. Are there clear research questions?	Yes	Yes	Yes	Yes	Yes	Yes
	S2. Do the collected data allow the study to address the research questions?	Yes	Yes	Yes	Yes	Yes	Yes
2. Quantitative randomized controlled trials	2.1. Is randomization appropriately performed?	Yes	Cannot tell	Yes	Cannot tell	Yes	Cannot tell
	2.2. Are the groups comparable at baseline?	Yes	No	Yes	Yes	Yes	Yes
	2.3. Are there complete outcome data?	Yes	Yes	Yes	Yes	Yes	Yes
	2.4. Are the outcome assessors blinded to the intervention provided?	Yes	Yes	Yes	Cannot tell	Yes	Cannot
	2.5. Did the participants adhere to the assigned intervention?	Yes	Yes	Yes	Yes	Yes	Yes

**Table 3 behavsci-15-01069-t003:** Study descriptions.

Authors, Region, and Study Design	Total Sample Size	Intervention Group	Control Group	Frequency and Duration of Intervention	Content of Intervention		Assessment	Main Results
					Experimental Group	Control Group		
[15] ([15]) UK RCT	68	21 (CST plus carer training) 10 F/11 M Age 75.4 ± 5.56 24 (CST) 9 F/15 M Age 76.8 ± 6.62	23 13 F/10 M Age 77.8 ± 7.47	45 min One time a week 14 weeks	Standardized CST manual with an RO board	Waitlist	MMSE ADAS-Cog QoL-AD QCPR	No changes in cognition (MMSE), quality of life (QoL-AD), or quality of carer–patient relationship (QCPR) over time and no significant differences between groups at follow-up. Significant decline in cognition between baseline and follow-up as assessed by ADAS-Cog, but no differences between groups.
[32] ([32]) Hong Kong RCT	176 150 F/26 M Age 75.41 ± 7.31 Years of Education 3.54 ± 3.77	86 75 F 7 11 M Age 77.41 ± 6.75 Years of education 2.92 ± 3.36	90 75 F/15 M Age 73.50 ± 7.35 Years of Education 4.17 ± 4.05	60 min One time a week 8 weeks	Active Mind, CST version targeted to Chinese culture	Treatment as usual	CDRS MMSE Cantonese version SF12	Improvements in cognition (CDRS) and QoL (SF12) in the IG. Both the IG and the CG showed improvement after intervention, but this was more prominent in the IG.
[41] ([41]) Germany RCT	71 60 F/11 M Age 86.37 ± 4.45	36 30 F/6 M Age 86.25 ± 4.76	35 30 F/5 M Age 86.49 ± 4.17	60 min Two times a week 8 weeks	NEUROvitalis Sinnreich	Usual care	ADAS-Cog QoL NPI-NH ADCS-ADL	No significant interaction effects regarding Time x Group. Significant within-subject effect regarding Time (pre-test to follow-up) for QoL and ADL scale, indicating that both worsened.
[45] ([45]) Sweden RCT	30 couples	Collaborative Intervention: 10 7 F/3 M Age 74.4 ± 6.0 Individual Intervention: 10 4 F/6 M Age 74.8 ± 6.7	10 4 F/6 M Age 77.0 ± 6.6	60 min One time a week 8 weeks	Cognitive training	Did not receive any intervention	Objective recall random/clustered recall of non-categorizable words ZBI BDI Collaborative object recall random Health Questionnaire MMSE Digit Span forward and backward WAIS-R Verbal fluency task Digit Symbol WAIS-R Verbal Ability Swedish Synonym Test	People with dementia in the collaborative group improved their memory performance from pre-test to post-test compared to the two other groups. No improvements in collaborative memory performance as a function of training. Separately, caregivers showed a reliable decrease in recall performance from pre-test to post-test, whereas their spouses with dementia showed an improvement in object recall. No changes in measures of reported depressive symptoms or in perceived caregiving burden for the caregivers as a function of the intervention; however, there was an increase in depression scores for all groups, which may reflect a response to disease progression.
[52] ([52]) Sub-Saharan Africa Stepped-wedge design	34 29 F/5 M Age 80.0 Years of Education 10	Immediate Start Group 1: 8 8 F/0 M Age 84.0 Years of Education 1 Group 2: 8 6 F/2 M Age 80.0 Years of Education 7	Delayed Start Group 3: 8 5 F/3 M Age 83.5 Years of Education 1 Group 4: 10 10 F/0 M Age 80.0 Years of Education 1	Two times a week 7 weeks	CST-SSA (CST adapted version for sub-Saharan Africa)	Delayed start groups acted as control	WHOQOL-Brief WHODAS 2.0 ADAS-Cog HADS NPI ZBI	Significant improvements in cognition (ADAS-Cog) and in the physical health domain of the WHOQOL-Bref. According to the caregivers, there were significant improvements in symptoms of anxiety and the number and severity of BPS and distress caused by BPS of dementia in the person they cared for, as assessed by the NPI.
[56] ([56]) USA RCT	78 27 F/51 M Age 73.6 ± 8.0 Years of Education 12.6 ± 4.1	25	28 placebo group 25 control group	60 min One time a day 6 days	Active cognitive stimulation program	Placebo: activities similar to those in the experimental group Control: wait-list	DRS WMS-R F-A-S test Geriatric Coping Schedule Visual Memory Span Digit Span Memory and Behavior Problems Checklist	Care recipients in the IG had improvements post-treatment in overall cognitive functioning (word fluency, recall of non-verbal material). There was a tendency in all outcomes for the IG to regress toward baseline by the ninth month. Instead, the CG declined post-treatment and at the 9-month follow-up; the placebo group remained at baseline maintenance level over time.

**Table 4 behavsci-15-01069-t004:** Cognitive and non-cognitive factors associated with greater benefit from cognitive stimulation.

	Demographic Factors	Cognitive Factors	Emotional Factors	Social Factors	Quality of Life
[15] ([15])		Baseline level of cognitive functioning			
[32] ([32])	Education level		Baseline level of depressive symptoms		Baseline level of quality of life
[41] ([41])		Baseline level of cognitive functioning			
[45] ([45])				Active participation of caregiver during CS session	
[52] ([52])	Education level				
[56] ([56])	Gender				

## Data Availability

No new data were created or analyzed in this study. Data sharing is not applicable to this article.

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
