# Peer review of "Cognitive and Non-Cognitive Predictors of Response to Cognitive Stimulation Interventions in Dementia: A Systematic Review Aiming for Personalization"

_behavsci, 2025, doi:10.3390/bs15081069_

Round 1

Reviewer 1 Report

Comments and Suggestions for Authors

Thank you for the invitation to review this article. The author conducted a systematic review that investigated cognitive and non-cognitive aspects of people with dementia that predict response or effectiveness of cognitive stimulation interventions. The results provide insights into the factors that are associated with a benefit from these interventions.

The systematic review is well conducted, and the manuscript is very well written. I congratulate the authors on their work. The introduction is well written and so are the other sections. The discussion is sufficient with satisfactory comparison to previous studies.

One minor comment: the limitations and strengths can be moved before conclusions.  

Author Response

COMMENT 1: Thank you for the invitation to review this article. The author conducted a systematic review that investigated cognitive and non-cognitive aspects of people with dementia that predict response or effectiveness of cognitive stimulation interventions. The results provide insights into the factors that are associated with a benefit from these interventions. 

The systematic review is well conducted, and the manuscript is very well written. I congratulate the authors on their work. The introduction is well written and so are the other sections. The discussion is sufficient with satisfactory comparison to previous studies. 

One minor comment: the limitations and strengths can be moved before conclusions.  

RESPONSE 1: Dear reviewer, Thank you very much for your thoughtful and constructive feedback, as well as for your kind words regarding the quality of our work. We truly appreciate your positive assessment of the review's methodological rigor, the clarity of the manuscript, and the relevance of our findings. 

In response to your suggestion, we have revised the structure of the manuscript in the new version by moving the "Limitations and Strengths" section to precede the "Conclusions," in accordance with your recommendation (please, see p. 17, R500-538) 

Thank you again for your valuable insights and generous comments. 

Reviewer 2 Report

Comments and Suggestions for Authors

This systematic review addresses an important and timely question regarding individual factors that influence the effectiveness of cognitive stimulation therapy (CST) in people with dementia. While the topic is clinically relevant, the manuscript suffers from methodological limitations, redundancy in the introduction, and limited interpretive depth given the small number of included studies. There are several key points the authors need to address to improve the quality of this manuscript.

Major:

  1. Only 6 studies were included—this is surprisingly low given the volume of CST research.
  2. In the introduction, the statements are redundant, with similar ideas repeated in multiple places. For example, CST’s effectiveness and endorsement is repeated across different paragraphs with minimal added value. Statements in R50–54, R66–77, and R78–79 all reinforce CST’s value but cite overlapping studies. The authors should consolidate them into one paragraph.overlapping studies. Authors should consolidate them into one graph.
  3. R87–92: This was already implied in previous paragraphs about inconsistent effects across outcomes. It adds little new insight, and the same idea could be introduced just once when transitioning into the review rationale.
  4. In introduction, authors also need to consolidate repeated points on variability in CST response.
  5. Only one reviewer extracted data and assessed the risk of bias, while others only screened. There should be two independent reviewers for data extraction and risk of bias to reduce error and bias.
  6. The inclusion/exclusion criteria is overly restrictive, as reflected in the very small final number (n=6). This can introduce selection bias and reduce generalizability.
  7. In the results section, long, dense paragraphs with complex sentences reduce readability. The authors should improve this.
  8. Studies differed in CST content and delivery, outcome measures, cultural context, and intervention duration and dose. However, these differences are not systematically discussed.

Minor:

  1. There are Several grammar issues and awkward phrases. For example, R93“Cognitive and non-cognitive aspects ensuring improvement” is awkward; R30 “Consequentially” should be “Consequently”, R84 “Ambiguous evidences” should be “ambiguous evidence”.
  2. Table 1 says "Cognitive Stimulation Therapy or cognitive stimulation interventions" which is unclear if they only mean CST (structured, manualized) or any CS intervention.

Author Response

REVIEWER 2 

This systematic review addresses an important and timely question regarding individual factors that influence the effectiveness of cognitive stimulation therapy (CST) in people with dementia. While the topic is clinically relevant, the manuscript suffers from methodological limitations, redundancy in the introduction, and limited interpretive depth given the small number of included studies. There are several key points the authors need to address to improve the quality of this manuscript. 

We sincerely thank the reviewer for their thoughtful and constructive feedback, as well as for acknowledging the relevance and clinical importance of the topic we addressed. We greatly appreciate the time and effort dedicated to evaluating our work. We carefully considered all the concerns raised regarding methodological aspects, redundancy in the introduction, and the limited interpretive scope due to the number of included studies. We have done our best to address each of these points through substantial revisions to the manuscript — including a clearer rationale for our methodological choices, a streamlined and more focused introduction, and a more nuanced discussion of the findings in light of current literature. We hope that the changes we have made — guided directly by the reviewer’s valuable suggestions — have improved the overall quality and clarity of the manuscript. We remain grateful for the opportunity to revise our work and are hopeful that it now better meets the expectations of the reviewers and the journal. 

Major: 

COMMENT 1: Only 6 studies were included—this is surprisingly low given the volume of CST research. 

RESPONSE 1: Dear Reviewer, thank you for your constructive feedback. We recognize that including only six studies may appear surprisingly low, particularly given the wide volume of CST research. However, the primary aim of our review was not to assess the overall efficacy of cognitive stimulation therapy, but rather to investigate specific predictive factors—namely demographic, cognitive, and non-cognitive characteristics that influence individual response to cognitive stimulation interventions. 

To address this focused question with methodological rigor, we applied strict eligibility criteria, restricting the review to randomized controlled trials that explicitly reported at least one predictor variable related to intervention outcomes. As a consequence, many well-designed CST studies were excluded because they did not explore predictors of effectiveness, which was the crux of our research question. 

We acknowledge that the small number of included studies represents a limitation. Nonetheless, established guidance affirms that methodological precision and relevance often outweigh the sheer quantity of included studies—especially in reviews addressing emerging or specialized topics. As stated in the Cochrane Handbook, a systematic review can be methodologically sound even with a small number of studies, provided that inclusion criteria are justified and aligned with the research question. Moreover, we refer reviewers to our Comment response #6, which details and clarifies our eligibility criteria and rationale in full. We hope this explanation clarifies the rationale behind our selection process. We have also clarified this aspect in the “Limitations and Strengths” section of the manuscript, p. 18 R526-528 (“This focused scope, while limiting the number of included studies, was intentional and aligned with the specific aim of identifying predictive factors, rather than evaluating general efficacy or to maximize generalizability”). Please let us know if further clarification is needed. 

COMMENT 2. In the introduction, the statements are redundant, with similar ideas repeated in multiple places. For example, CST’s effectiveness and endorsement is repeated across different paragraphs with minimal added value. Statements in R50–54, R66–77, and R78–79 all reinforce CST’s value but cite overlapping studies. The authors should consolidate them into one paragraph. 

RESPONSE 2: We thank the reviewer for this thoughtful comment. We agree that the original version of the “Introduction” contained redundant statements regarding the effectiveness and clinical endorsement of Cognitive Stimulation Therapy (CST). As suggested, we have now consolidated the content from R50–54, R66–77, and R78–79 in the previous manuscript into a single, concise paragraph. 

The revised version maintains all key information, including the rationale for psychosocial approaches, the evidence base and multidimensional benefits of CST, and its endorsement by both UK and Italian national guidelines. We also revised the references to eliminate overlaps and improve clarity. At the same time, we made sure to preserve the original logical progression of the paragraph—from general psychosocial approaches to cognitive stimulation, to CST—to maintain conceptual clarity for the reader. The new paragraph can be found on page 2, R55-61 and R64-77 of the revised manuscript. 

R55-61 “Psychosocial approaches represent a valuable non-pharmacological alternative for managing cognitive decline, behavioral symptoms, and overall quality of life in people with dementia (Sikkes et al., 2021). Among the various psychosocial interventions—such as behavioral, environmental, and cognitive approaches—cognitive stimulation (CS) has shown the most consistent benefits in people with mild to moderate dementia (Woods et al., 2012; McDermott et al., 2019).”  

R64-77 “Within the broad category of CS programs, Cognitive Stimulation Therapy (CST) stands out as a structured and manualized group-based intervention specifically developed for this population (Spector et al., 2003). CST is supported by robust evidence demonstrating positive effects not only on global cognition, but also on specific cognitive domains such as memory, attention, language, orientation, neuropsychiatric symptoms, daily functioning, and quality of life (Aguirre et al., 2013; Kim et al., 2016; Gomez-Soria et al., 2023; Xiang et al., 2024). For these reasons, CST is currently the only psychosocial intervention formally recommended to improve cognition in people with dementia by both the UK’s National Institute for Health and Care Excellence (NICE, 2018) and the Italian national guidelines (ISS-SNLG, 2024). It is also widely considered a cost-effective gold-standard intervention in non-pharmacological dementia care (Knapp et al., 2006; McDermott et al., 2019).

COMMENT 3:R87–92: This was already implied in previous paragraphs about inconsistent effects across outcomes. It adds little new insight, and the same idea could be introduced just once when transitioning into the review rationale. 

RESPONSE 3: We thank the reviewer for this observation. We acknowledge that the sentence in R87–92 in the previous manuscript version repeated a concept already mentioned earlier—that is, the inconsistent findings across studies on the effects of cognitive stimulation interventions. To improve clarity and avoid redundancy, we have removed the repeated statement and ensured that the idea is introduced only once, specifically in the transition to the rationale of the review. 

This change helps maintain a more concise and focused introduction, while still highlighting the variability in study outcomes as a justification for our systematic review. Please, see p. 2, R77-85. 

Despite the growing body of literature supporting the effectiveness of cognitive stimulation interventions, findings remain partially inconsistent across studies. For example, Leroi et al. (2019) reported non-significant effects on cognitive function, while Cafferata et al. (2021) found ambiguous results regarding activities of daily living, and Wong et al. (2021) highlighted mixed evidence on depressive symptoms and quality of life. One possible explanation for this variability may lie in the individual characteristics of people with dementia, including both cognitive and non-cognitive factors, which may influence the outcomes of such interventions.”  

COMMENT 4: In introduction, authors also need to consolidate repeated points on variability in CST response. 

RESPONSE 4: We thank the reviewer for this valuable observation. We acknowledge that the variability in CST outcomes was previously mentioned across multiple sections of the introduction. In line with the suggestion, we have now consolidated these points into a single, more concise and coherent paragraph (see p. 2, lines 77–85). This revised paragraph integrates both the inconsistencies reported in previous studies and the rationale of our review, focusing on the influence of individual-level cognitive and non-cognitive characteristics on CST outcomes. In doing so, we avoided redundancy while maintaining the relevance of the issue, as already addressed in our response to the previous comment (please see response to previous comment n.3) 

COMMENT 5: Only one reviewer extracted data and assessed the risk of bias, while others only screened. There should be two independent reviewers for data extraction and risk of bias to reduce error and bias. 

RESPONSE 5: We thank the reviewer for this comment and the opportunity to clarify. As for the risk of bias assessment, we would like to point out that this process was carried out by one author (L.F.) and independently reviewed by a second author (M.Q.), with a third author (R.C.) available in case of disagreement. This procedure, as described in the manuscript (see p. 6, R 253–255 “The risk of bias assessment was conducted by one author (L.F.) and reviewed by a second author (M.Q.); possible differences in the assessment would have been checked by a third reviewer (R.C.)”), ensured a reliable and transparent assessment involving two reviewers, consistent with current methodological standards. Moreover, this approach aligns with the recommendations provided in the MMAT “Criteria Manual,” which advises that, since critical appraisal involves judgment making, it is recommended to have at least two reviewers independently involved in the appraisal process. Additionally, the manual suggests that users may be assisted by a colleague with specific expertise when needed—hence the involvement of a third author to resolve disagreements and provide expert input when necessary (please, see Hong, Q.N., Pluye, P., Fabregues, S., Bartlett, G., Boardman, F., Cargo, M., Dagenais, P., Gagnon, M.P., Griffiths, F., Nicolau, B., O’Cathain, A., Rousseau, M.C., Vedel, I. (2018). Mixed Methods Appraisal Tool (MMAT), version 2018. Registration of Copiright (#1148552), Canadian Intellectual Property Office, Industry Canada.) 

Regarding data extraction, we acknowledge that this task was initially performed by one author. However, the extracted data were then carefully cross-checked and discussed with the full research team, who were all thoroughly familiar with the six included studies. While this does not equate to a fully independent double-extraction process, we believe that the collaborative verification phase effectively minimized the risk of error or bias. We have now added a sentence to clarify this aspect in the manuscript (see p. 6, R 240–243): “However, all extracted information was subsequentially reviewed by all co-authors (G.D., M.Q., L.C., M.B., S.T., A.A., I.C.), who were thoroughly familiar with the included studies. This collaborative process allowed for cross-checking of data and minimization of potential errors or biases.” 

COMMENT 6: The inclusion/exclusion criteria is overly restrictive, as reflected in the very small final number (n=6). This can introduce selection bias and reduce generalizability. 

RESPONSE 6: We appreciate the reviewer’s concern regarding the strict inclusion/exclusion criteria, which indeed resulted in a small final number of included studies (n=6), However, our methodological choices were guided by the aim to ensure a high level of rigor and consistency across included studies. Below, we provide a detailed justification for each criterion: 

  1. Participants: as inclusion criteria, we included only studies with a population of adults aged 65 and above with a diagnosis of mild to moderate dementia. This restriction was necessary to ensure homogeneity of the target population, as cognitive stimulation interventions may have different effects depending on age and dementia severity (Spector et al., 2003; Woods et al., 2012). From a clinical and epidemiological standpoint, dementia occurring before the age of 65 is often classified as Young-Onset Dementia (YOD) (Rossor et al., 2010), also known as Early-Onset Dementia (EOD) (Johannessen & Moller, 2011), which may involve different etiologies, care needs, and psychosocial implications compared to late-onset dementia (Harvey et al., 2003; van Vliet et al., 2010). So, including younger individuals would therefore have introduced significant heterogeneity and potentially confounded our analysis. We also restricted inclusion to individuals with mild to moderate dementia, as cognitive stimulation interventions are specifically recommended for this subgroup. Evidence from both clinical trials and international guidelines indicates that cognitive stimulation interventions is most effective – and most appropriate – for people with mild to moderate levels of cognitive impairment (Spector et al., 2003; NICE, 2018). In people with severe dementia, cognitive stimulation interventions may be less feasible or have diminished efficacy due to greater functional limitations and lower cognitive reserve. Likewise, as exclusion criteria we excluded studies focusing on individuals younger than 65 years, individuals without a formal diagnosis of dementia, or those presenting with other medical or psychiatric conditions such as major psychiatric disorders, stroke, or traumatic brain injury. This last decision was grounded in standard diagnostic criteria for dementia. DSM-5 and ICD-10 require that cognitive decline not be better explained by other neurological, psychiatric or systemic medical disorders (APA, 2013; WHO, 1992). Our goal was to focus exclusively on studies involving individuals with a confirmed diagnosis of dementia as the primary condition, in line with the core criteria.  
  1. Intervention: we focused exclusively on structured cognitive stimulation interventions that fit the conceptual definition proposed by Clare and Woods (2004), namely, “engagement in a range of activities and discussions aimed at the general enhancement of cognitive and social functioning”, rather than interventions targeting a specific cognitive function. This type of multi-domain, non-specific stimulation is distinct from cognitive training (which targets specific domains such as memory or attention), cognitive rehabilitation (which is goal-oriented and individualized), and other psychosocial interventions (Clare & Woods, 2004; Spector et al., 2003; Woods et al., 2012). The decision to exclude studies focusing on cognitive stimulation targeting a single cognitive domain is supported by existing literature emphasizing the superior efficacy of multi-domain cognitive stimulation interventions. Clare and Woods (2004) clarify that cognitive stimulation is characterized by engaging multiple cognitive domains, distinguishing it from domain-specific cognitive training or rehabilitation. Systematic reviews, including those by Woods et al. (2012) and Bahar-Fuchs et al. (2019), provide evidence that multi-domain cognitive stimulation leads to broader improvements in cognitive functioning and daily living activities compared to interventions focusing on a single cognitive domain. Furthermore, clinical guidelines such as those from NICE (2018) recommend multi-domain approaches as standard practice for cognitive interventions in dementia care, reinforcing the rationale behind our exclusion criteria. Based on this rationale, we excluded studies that: 
  • Focused on isolated stimulation of a single cognitive domain (e.g., memory-only tasks), 
  • Implemented multifactorial interventions (e.g. combining physical activity, diet, and cognitive task without isolating the cognitive stimulation component); 
  • Applied other psychosocial interventions (e.g., reminiscence therapy or Reality Orientation Therapy ROT) that do not involve structured cognitive stimulation. 
  • Or combined cognitive stimulation interventions with pharmacological treatments, as this would have made it difficult to isolate the effect of the cognitive component. 
  1. Control: we included only studies that used a passive control group, defined as participants receiving standard care (treatment as usual) or no active intervention. This choice was made to ensure that any observed effects could be more confidently attributed to the cognitive stimulation intervention, minimizing potential confounding effects introduced by other simultaneous activities. Accordingly, we excluded studies with an active control group, such as those engaging participants in alternative cognitive, social, or behavoiral activities (e.g., recreational groups, psychoeducation, or other non –specific engagement strategies). Active controls, while useful in some contexts, may reduce the ability to isolate the unique contribution of cognitive stimulation, particularly in a systematic review aiming to explore moderators of response. 
  1. Outcomes: eligible studies were required to evaluate multiple outcome domains – including cognitive, functional psychological, or affective measures – assessed both before and after the cognitive stimulation intervention. We specifically included studies that evaluated the effects of cognitive stimulation on multiple cognitive domains rather than on a single cognitive function because dementia typically affects a range of cognitive abilities simultaneously. Measuring improvement across multiple domains allows for a more comprehensive and ecologically valid assessment of the intervention’s effectiveness. This approach aligns with existing literature emphasizing the importance of multi-domain cognitive assessments in dementia research to detect meaningful and generalizable changes (Clare & Woods, 2004; Yates et al., 2016). Therefore, our inclusion criteria aimed to capture studies that evaluate cognitive outcomes more comprehensively to inform the development of personalized and effective cognitive stimulation interventions. Furthermore, a key inclusion criterion was that studies had to explore the influence of individual cognitive and/or non-cognitive characteristics (e.g. baseline cognitive level, mood, education, age) on the outcomes of the intervention. This focus reflects the increasing recognition of the person-centered approaches in dementia care and research, which emphasize the importance of identifying which individuals are more likely to benefit from specific interventions (Clare et al., 2019; Yates et al., 2016). Rather than simply documenting average effects, our review sought to summarize findings on moderators of intervention response – an area that remains relatively underexplored in the current literature. Accordingly, we excluded studies that: 
  • Did not report pre- and post-interventions outcome measures; 
  • Did not include cognitive and non-cognitive outcomes; 
  • Focused solely on the improvement of a single cognitive function; 
  • Or did not explore the relationship between individual characteristics and intervention outcomes (e.g., studies reporting only overall group-level effects, without analysis of influencing factors). 
  1. Study design: we included only randomized controlled trials (RCTs), which are widely considered the gold standard for evaluating the efficacy of interventions and for minimizing selection bias and confounding (Moher et al., 2009). This choice was made to ensure that the evidence reviewed was based on robust and methodologically sound designs capable of supporting causal inferences.  A part of this approach, we excluded: 
  • Studies without a control group. 
  • Studies that used only a pre-post comparison within a single group, which do not allow for attribution of effects to the intervention with a sufficient confidence. 

We also excluded protocol papers, abstracts or posters from congress, and studies for which the full text was not available. 

We acknowledge that our inclusion and exclusion criteria were restrictive and led to the selection of a small number of studies. However, as detailed above, each criterion was carefully defined based on theoretical and methodological considerations, aligned with our specific research objective: to review studies evaluating the influence of individual cognitive and non-cognitive characteristics on the outcome of multi-domain cognitive stimulation interventions in people with mild to moderate dementia. These methodological choices – particularly the focus on multi-domain cognitive stimulation, the requirement for pre-post outcome measurements, and especially the inclusion of only randomized controlled trials (RCTs) - were necessary to ensure consistency, conceptual clarity, and analytical relevance. As a result, we inevitably excluded a substantial number of studies on cognitive stimulation that did not meet these criteria, despite their potential interest from a broader perspective. Above all, the restrictive criteria concerning the outcome and the type of study design led us to eliminate a large number of studies, as can be seen from the Prism Flow Chart (in which we have indicated the number of studies eliminated for each reason): many studies were not RCTs, and a lot of did not indicate cognitive or non-cognitive factors that could have influenced the effectiveness of the intervention. Importantly, our aim was not to produce generalizable conclusions about the overall efficacy of cognitive stimulation, but rather to synthesize and critically appraise the existing literature regarding the individual cognitive and non-cognitive characteristics that may predict differential response to such interventions in people with mild to moderate dementia. 

This part of our manuscript has been carefully revised, and the rationale for our inclusion and exclusion criteria were clarified and incorporated into the list of the criteria. Please, see p. 4-5, R137-229:  

(a) studies that included participants older than 65 years of age and with a diagnosis of mild to moderate dementia. From a clinical and epidemiological standpoint, dementia occurring before the age of 65 is often classified as Young-Onset Dementia (YOD) (Rossor et al., 2010), also known as Early-Onset Dementia (EOD) (Johannessen & Moller, 2013), which may involve different etiologies, care needs, and psychosocial implications compared to late-onset dementia (Harvey et al., 2003; van Vliet et al., 2010). So, including younger individuals would therefore have introduced significant heterogeneity and potentially confounded our analysis. We also restricted inclusion to individuals with mild to moderate dementia, as cognitive stimulation interventions are specifically recommended for this subgroup. Evidence from both clinical trials and international guidelines indicates that cognitive stimulation interventions is most effective – and most appropriate – for people with mild to moderate levels of cognitive impairment (Spector et al., 2003; NICE, 2018). In people with severe dementia, cognitive stimulation interventions may be less feasible or have diminished efficacy due to greater functional limitations and lower cognitive reserve.  

(b) studies that included isolated cognitive stimulation treatment, according to Clare and Woods definition of cognitive stimulation as “engagement in a range of group activities and discussions aimed at general enhancement of cognitive and social functioning”, rather than interventions targeting a specific cognitive function. This type of multi-domain, non-specific stimulation is distinct from cognitive training (which targets specific domains such as memory or attention), cognitive rehabilitation (which is goal-oriented and individualized), and other psychosocial interventions (Clare & Woods, 2004; Spector et al., 2003; Woods et al., 2012). This category includes both standardized Cognitive Stimulation Therapy (CST) protocols as well as other cognitive stimulation interventions sharing similar principles but not strictly following the manualized CST. 

(c) studies that included a passive control group that received standard care (treatment as usual) or no active treatment. This choice was made to ensure that any observed effects could be more confidently attributed to the cognitive stimulation intervention, minimizing potential confounding effects introduced by other simultaneous activities. 

(d) studies which evaluated multiple outcome domains – including functional, cognitive, psychological and affective outcomes – assessed both before and after cognitive stimulation intervention and that underlined the influence of cognitive and non-cognitive aspects of people with dementia on the gain resulting from cognitive stimulation intervention. We specifically included studies that evaluated the effects of cognitive stimulation on multiple cognitive domains rather than on a single cognitive function because dementia typically affects a range of cognitive abilities simultaneously. Measuring improvement across multiple domains allows for a more comprehensive and ecologically valid assessment of the intervention’s effectiveness. This approach aligns with existing literature emphasizing the importance of multi-domain cognitive assessments in dementia research to detect meaningful and generalizable changes (Clare & Woods, 2004; Yates et al., 2016). Therefore, our inclusion criteria aimed to capture studies that evaluate cognitive outcomes more comprehensively to inform the development of personalized and effective cognitive stimulation interventions. Furthermore, a key inclusion criterion was that studies had to explore the influence of individual cognitive and/or non-cognitive characteristics (e.g. baseline cognitive level, mood, education, age) on the outcomes of the intervention. This focus reflects the increasing recognition of the person-centered approaches in dementia care and research, which emphasize the importance of identifying which individuals are more likely to benefit from specific interventions (Clare et al., 2019; Yates et al., 2016). 

(e) study design was a randomized controlled trials (RCTs) to select studies with demonstrated evidence of efficacy with higher standards. RCTs are widely considered the gold standard for evaluating the efficacy of interventions and for minimizing selection bias and confounding (Moher et al., 2009). This choice was made to ensure that the evidence reviewed was based on robust and methodologically sound designs capable of supporting causal inferences. 

2.3. Exclusion criteria 

Exclusion criteria were:  

  1. a) studies focusing on adult population younger than 65 years, or without a diagnosis of dementia, or those presenting with other medical or psychiatric conditions such as major psychiatric disorders, stroke, or traumatic brain injury. This last decision was grounded in standard diagnostic criteria for dementia. DSM-5 and ICD-10 require that cognitive decline not be better explained by other neurological, psychiatric or systemic medical disorders (APA, 2013; WHO, 1992).
  2. b) studies that included isolated cognitive stimulation interventions focused on a single cognitive function (e.g., memory-only tasks), multifactorial intervention (e.g. combining physical activity, diet, and cognitive task without isolating the cognitive stimulation component), other psychosocial interventions (e.g., reminiscence therapy or Reality Orientation Therapy ROT) that do not involve cognitive stimulation, or combined cognitive stimulation interventions with pharmacological treatments, as this would have made it difficult to isolate the effect of the cognitive component. The decision to exclude studies focusing on cognitive stimulation targeting a single cognitive domain is supported by existing literature emphasizing the superior efficacy of multi-domain cognitive stimulation interventions. Clare and Woods (2004) clarify that cognitive stimulation is characterized by engaging multiple cognitive domains, distinguishing it from domain-specific cognitive training or rehabilitation. Systematic reviews, including those by Woods et al. (2012) and Bahar-Fuchs et al. (2019), provide evidence that multi-domain cognitive stimulation leads to broader improvements in cognitive functioning and daily living activities compared to interventions focusing on a single cognitive domain. Furthermore, clinical guidelines such as those from NICE (2018) recommend multi-domain approaches as standard practice for cognitive interventions in dementia care, reinforcing the rationale behind our exclusion criteria.
  3. c) studies with an active control group, such as those engaging participants in alternative cognitive, social, or behavioural activities (e.g., recreational groups, psychoeducation, or other non –specific engagement strategies). Active controls, while useful in some contexts, may reduce the ability to isolate the unique contribution of cognitive stimulation, particularly in a systematic review aiming to explore moderators of response.  
  4. d) studies that did not include cognitive and non-cognitive outcomes or that focused solely on the improvement of a single cognitive function or did not explore the relationship between individual characteristics and intervention outcomes (e.g., studies reporting only overall group-level effects, without analysis of influencing factors).
  5. e) studies without a control group or with only a pre and post treatment comparison within a single group. Furthermore, protocol studies, abstracts or posters from congress, and studies with no full text available were excluded.

COMMENT 7: In the results section, long, dense paragraphs with complex sentences reduce readability. The authors should improve this. 

RESPONSE 7: We thank the reviewer for this important observation. In response, we have revised the Results section to improve clarity and readability. Specifically, we broke down long paragraphs into shorter, more focused ones and restructured complex sentences for better flow, aiming to enhance the accessibility and comprehensibility of the findings. We believe these changes substantially improve the presentation of the results. Please, see p.6-14. R256-379. 

COMMENT 8: Studies differed in CST content and delivery, outcome measures, cultural context, and intervention duration and dose. However, these differences are not systematically discussed. 

RESPONSE 8: We thank the Reviewer for this important observation. In the revised manuscript, we have added a dedicated paragraph in the Discussion section (see p. 16-17, lines 434–464) that systematically addresses the heterogeneity across the included studies. Below, we provide a point-by-point justification of these differences, supported by relevant literature: 

  1. Setting differences: although the included studies were conducted in different settings—such as community centers, care homes, or at-home environments—evidence suggests that setting does not significantly influence the effectiveness of CST. The Cochrane systematic review by Woods et al. (2018) found no significant subgroup differences in outcomes based on setting (e.g., community vs care home), supporting the generalizability of CST outcomes across different environments.
  2. Cultural context: while our review included studies conducted in culturally diverse regions, prior literature demonstrates that CST is effective across multiple cultures. Notably, the same Cochrane review by Woods et al. (2018) included 33 randomized controlled trials conducted across 17 countries on five continents and found consistent benefits of CST on cognition and quality of life, regardless of cultural or regional variations. Furthermore, structured adaptation work supports the cross-cultural applicability of CST. Aguirre et al. (2014) proposed the FMAP (Formative Method for Adapting Psychotherapy) to guide culturally sensitive adaptations of CST in countries such as Tanzania, Nigeria, China, South Asia, and Japan, while preserving its structural integrity. Pilot studies implementing culturally adapted CST programs in Hong Kong (Wong et al., 2018), Portugal (Alvares Pereira et al., 2022), and among Māori communities in New Zealand (Dudley et al., 2025) report consistent improvements in cognition and quality of life. These findings reinforce the external validity of CST across culturally diverse populations.
  3. Outcome measure heterogeneity: We acknowledge the use of different outcome measures across the studies. However, we note that this variability does not reflect conceptual inconsistency. All included studies employed at least one standardized global cognitive screening tool (e.g., MMSE or ADAS-Cog), and several included domain-specific cognitive assessments. Most studies also included at least one non-cognitive outcome, with quality of life being the most commonly assessed domain. Although the tools varied in format or cultural adaptation, they all targeted the same underlying constructs.
  4. Intervention duration and dose: the included studies differed somewhat in the number and frequency of CST sessions. However, this variability reflects practical adaptations rather than methodological limitations. The original CST protocol recommends 14 sessions over 7 weeks (twice weekly) (Spector et al., 2003), but research shows that CST retains its effectiveness across varied delivery formats.Recent meta-analyses (Chen et al., 2022; Sun et al., 2022) confirm that CST remains effective even when session duration, frequency, or total number of sessions slightly deviate from the original protocol. Therefore, while these differences exist, they do not undermine the validity of our findings, but rather demonstrate CST’s adaptability in real-world contexts. 

We thank you for your helpful suggestion. In accordance with your recommendation, we have incorporated a more detailed discussion of these topics in the “Discussion” section (please, see p.17, R478-508). 

Despite some heterogeneity among the included studies in terms of CST delivery, outcome measures, cultural context, and intervention duration and dose, these differences do not appear to undermine the consistency of the findings. Regarding the intervention setting, previous literature has shown that the effectiveness of CST is not significantly influenced by whether it is delivered in community-based environments, care homes, or at home. The Cochrane review by Woods et al. (2018) found no significant subgroup differences in outcomes based on setting, supporting the generalizability of CST interventions across diverse environments. With respect to cultural context, although the studies included in this review were conducted in different countries, evidence from large-scale systematic reviews and culturally adapted CST programs suggests that the intervention maintains its efficacy across diverse populations. For example, Woods et al. (2018) analyzed 33 RCTs across 17 countries and five continents, reporting consistent cognitive and quality of life improvements. Furthermore, Aguirre et al. (2014) proposed the Formative Method for Adapting Psychotherapy (FMAP) to guide the cultural adaptation of CST in low- and middle-income countries, ensuring the intervention remains structurally consistent. Pilot studies from Hong Kong (Wong et al., 2018), Portugal (Alvares Pereira et al., 2022), and Māori communities in New Zealand (Dudley et al., 2025) further support the cross-cultural applicability of CST, with adaptations enhancing local acceptability without compromising core elements. Although different cognitive and non-cognitive outcome measures were used across the included studies, these tools were largely aimed at assessing the same domains—particularly global cognitive function (e.g., MMSE, ADAS-Cog), specific cognitive abilities, and quality of life. This suggests that, despite the diversity of instruments, the studies maintained conceptual consistency in their outcome assessment. Lastly, differences in CST duration and dose reflect real-world implementation rather than methodological inconsistency. While the original protocol suggests 14 sessions over seven weeks (Spector et al., 2003), recent meta-analyses (Chen et al., 2022; Sun et al., 2022) have confirmed that CST maintains its efficacy even when session number or frequency varies moderately. In fact, most of the included studies in our review provided interventions lasting approximately 60 minutes per session, and the number of sessions generally ranged from 8 to 14. This suggests that CST can be adapted flexibly in duration and format without a significant loss of efficacy.

Minor: 

COMMENT 9: There are Several grammar issues and awkward phrases. For example, R93“Cognitive and non-cognitive aspects ensuring improvement” is awkward; R30 “Consequentially” should be “Consequently”, R84 “Ambiguous evidences” should be “ambiguous evidence”.  

RESPONSE 9: Dear reviewer, thank you for pointing out these grammatical and syntactical issues. The sentence in R93 has been revised for improved clarity and flow (please, see p.2, R85-88: “This observation highlights the importance of further investigating which individual-level predictors may moderate or mediate the effectiveness of cognitive stimulation and led to the development of more personalized interventions”), and the grammatical errors in R31 have been corrected (please, see p. 1, R31). The sentence in R84 has been reformulated entirely to avoid ambiguity. We hope these revisions have improved the overall readability and quality of the manuscript. 

COMMENT 10: Table 1 says "Cognitive Stimulation Therapy or cognitive stimulation interventions" which is unclear if they only mean CST (structured, manualized) or any CS intervention. 

RESPONSE 10: Thank you for this important observation regarding the terminology used in Table 1. To clarify, our review included studies implementing cognitive stimulation interventions broadly, not limited to the standardized manualized CST protocol by Spector et al. This approach allowed us to capture a wider range of cognitive stimulation programs implemented in diverse settings and populations. 
Accordingly, we have revised Table 1 to specify “Cognitive Stimulation Interventions (including manualized CST protocols and other cognitive stimulation programs)”. Moreover, we have updated the inclusion criteria section to clearly reflect this broader scope of interventions (Please, see p. 4, R152-161: (b) studies that included isolated cognitive stimulation treatment, according to Clare and Woods definition of cognitive stimulation as “engagement in a range of group activities and discussions aimed at general enhancement of cognitive and social functioning”, rather than interventions targeting a specific cognitive function. This type of multi-domain, non-specific stimulation is distinct from cognitive training (which targets specific domains such as memory or attention), cognitive rehabilitation (which is goal-oriented and individualized), and other psychosocial interventions (Clare & Woods, 2004; Spector et al., 2003; Woods et al., 2012). This category includes both standardized Cognitive Stimulation Therapy (CST) protocols as well as other cognitive stimulation interventions sharing similar principles but not strictly following the manualized CST.”). This should prevent any confusion regarding the types of cognitive stimulation approaches considered in our review. We appreciate your helpful suggestion and believe these changes improve the clarity of our manuscript. 

Reviewer 3 Report

Comments and Suggestions for Authors

Comments to the Authors:

I appreciate the opportunity to review this article. This systematic review analysed 6 randomized controlled trials using cognitive stimulation therapy based on certain inclusion and exclusion criteria. It highlights the importance of considering individual characteristics, such as demographic, cognitive, and emotional factors, to optimize the benefits of cognitive stimulation interventions for people with dementia. Specifically, this article suggests that older females, individuals with lower education levels, and those with lower baseline cognitive function and low depressive symptoms tend to benefit more from these interventions.

I commend the authors for their detailed analysis and reporting. The manuscript is clearly written in professional, unambiguous language. However, I would suggest the following comments/typos that can be addressed before acceptance.

  1. The reference section needs extensive revision. There are a lot of wrongly inserted references. Also, some references cited in the text are not included. Please check all the references and revise.

Here are few examples for wrongly inserted ones: Reference# 1, 4, 5, 13, 15, 16, 20, 21, 23, 27, 29, 31, 38, 51, 59, 60, 70.

Please make sure that these references are included.

Line#67: Spector et al 2006

Line#359: Smith et al., 2013

Line#368: Orrell et al., 2017

Holopainen et al., 2019

Delete the repeated reference: Line #613-618: 57 and 58.

  1. Spector, A., Thorgrimsen, L., Woods, B., Royan, L., Davies, S., Butterworth, M., Orrell, M. (2003) Efficacy of an evidence- based cognitive stimulation therapy programme for people with dementia: randomised controlled trial. Br J Psychiatry 183, 248-254. 615
  2. Spector, A., Thorgrimsen, L., Woods, B., Royan, L., Davies, S., Butterworth, M., Orrell, M. (2003). Efficacy of an evidence- based cognitive stimulation therapy programme for people with dementia: randomised controlled trial. The British Journal of Psychiatry, 183(3), 248-254.

  1. Line#252-254: I would change the sentence “From the 6 included studies, factors that may influence the gain resulting from cognitive stimulation intervention were extracted a posteriori then assessed in demographic, cognitive, emotional, social factors and quality of life.”

to

“From the six included studies, factors that may influence the gain resulting from cognitive stimulation intervention were extracted a posteriori and then assessed in terms of demographic, cognitive, emotional, social factors, and quality of life.”

  1. Line#272: the experimental “groupp" with low baseline scores in cognitive performance (and fewer”

Change “groupp” to “group”.

  1. Line#334: According to Stern (2009) there is a point after which “reserve” can no longer with

I would change it to “According to Stern (2009) there is a point after which “cognitive-reserve” can no longer with.

  1. Line#344: Please correct the typo. “Another finding “in” that low baseline level of cognitive functions predicted improvements in cognition”.

“Another finding “is/was” that low baseline level of cognitive functions predicted improvements in cognition”. 

  1. Line#364-366: Please correct the typo. “It is also known that “conducing” the intervention in groups can encourage social interaction (Buschert et al., 2011; Olazaran et al., 2004) and increases the cognitive gain.”

“It is also known that “conducting” the intervention in groups can encourage social interaction (Buschert et al., 2011; Olazaran et al., 2004) and increases the cognitive gain.”

  1. Line#375-376: For clarity, please revise this sentence “Analogue results if it is considered another form of relationship, that is closer relationship or marital relationship.”

to

"Analogous results are observed when considering other types of relationships, such as a closer relationship or a marital relationship."

  1. The authors may consider deleting this repetition.

Line#358 : BPSD (behavioral and psychiatric symptoms of dementia) 

Line#420 behavioral and psychiatric symptoms of dementia (BPSD)

  1. Conclusion: Line#427

I have a concern regarding the generalization of findings related to the differential response to cognitive stimulation intervention based on demographic characteristics. Since the observation that women showed greater improvement and maintenance in memory performance over time compared to men is based on limited evidence (only one or two studies), I have a comment regarding the statement, 'Female older people with low formal education level, low baseline level of cognitive functions and lower depressive symptoms have greater improvements from cognitive stimulation intervention,'

I suggest rephrasing it to:

'Collectively, these RCT observations suggest that older female gender, individuals with low formal education level, low baseline level of cognitive functions, and lower depressive symptoms may have greater improvements from cognitive stimulation intervention.'

The could provide limited scope of the evidence and avoids overgeneralization.

Author Response

REVIEWER 3 

I appreciate the opportunity to review this article. This systematic review analysed 6 randomized controlled trials using cognitive stimulation therapy based on certain inclusion and exclusion criteria. It highlights the importance of considering individual characteristics, such as demographic, cognitive, and emotional factors, to optimize the benefits of cognitive stimulation interventions for people with dementia. Specifically, this article suggests that older females, individuals with lower education levels, and those with lower baseline cognitive function and low depressive symptoms tend to benefit more from these interventions. I commend the authors for their detailed analysis and reporting. The manuscript is clearly written in professional, unambiguous language. However, I would suggest the following comments/typos that can be addressed before acceptance. 

We sincerely thank the reviewer for their thoughtful and constructive comments. We appreciate your positive feedback on our manuscript, particularly regarding the clarity of the writing and the relevance of the findings. We are pleased that you found the analysis and reporting to be detailed and well-articulated. We have carefully addressed the specific comments and minor corrections you suggested, as outlined below (see responses to individual points). We believe that these revisions have helped to further improve the quality and clarity of the manuscript. 

COMMENT 1: The reference section needs extensive revision. There are a lot of wrongly inserted references. Also, some references cited in the text are not included. Please check all the references and revise. 

Here are few examples for wrongly inserted ones: Reference# 1, 4, 5, 13, 15, 16, 20, 21, 23, 27, 29, 31, 38, 51, 59, 60, 70. 

Please make sure that these references are included. 

Line#67: Spector et al 2006 

Line#359: Smith et al., 2013 

Line#368: Orrell et al., 2017 

Holopainen et al., 2019 

Delete the repeated reference: Line #613-618: 57 and 58. 

  1. Spector, A., Thorgrimsen, L., Woods, B., Royan, L., Davies, S., Butterworth, M., Orrell, M. (2003) Efficacy of an evidence- based cognitive stimulation therapy programme for people with dementia: randomised controlled trial. Br J Psychiatry 183, 248-254. 615 
  1. Spector, A., Thorgrimsen, L., Woods, B., Royan, L., Davies, S., Butterworth, M., Orrell, M. (2003). Efficacy of an evidence- based cognitive stimulation therapy programme for people with dementia: randomised controlled trial. The British Journal of Psychiatry, 183(3), 248-254. 

   

RESPONSE 1: Dear Reviewer, thank you very much for your careful and constructive feedback regarding the reference section. We greatly appreciate the time you dedicated to identifying inconsistencies, which helped us improve the accuracy and coherence of the manuscript. 

We have conducted a thorough revision of the reference list. Specifically: 

- References previously listed as #1, 4, 5, 13, 15, 23, 29, 38, and 51 have been deleted, as they are no longer cited in the final version of the manuscript. 

- Reference previously listed as #16 was initially mislabeled in the text as Smith et al., 2013; this has now been corrected to the appropriate source: Geda et al., 2013. [Geda, Y.E., Schneider, L.S., Gitlin, L.N., Miller, D.S., Smith, G.S., Bell, J., Evans, J., Lee, M., Porsteinsson, A., Lanctôt, K.L., Rosenberg, P.B., Sultzer, D.L., Francis, P.T., Brodaty, H., Padala, P.P., Onyike, C.U., Ortiz, L.A., Ancoli-Israel, S., Bliwise, D.L., Martin, J.L., Vitiello, M.V., Yaffe, K., Zee, P.C., Herrmann, N., Sweet, R.A., Ballard, C., Khin, N.A., Alfaro, C., Murray, P.S., Schultz, S., Lyketsos, C.G.; Neuropsychiatric Syndromes Professional Interest Area of ISTAART. (2013). Neuropsychiatric symptoms in Alzheimer's disease: Past progress and anticipation of the future. Alzheimer's & Dementia, 9(5), 602-608.] now reference #18. 

- Reference previously listed as #20 had an incorrect year; the correct reference is now Holopainen et al., 2017 [Holopainen, A., Siltanen, H., Okkonen, A. (2017). Factors Associated with the Quality of Life of People with Dementia and Quality of Life-Improving Interventions: Scoping Review. Dementia, 18.],  in the manuscript reference #24. 

- Reference previously listed as #21 refers to the MMAT tool [Hong, Q.N., Pluye, P., Fabregues, S., Bartlett, G., Boardman, F., Cargo, M., Dagenais, P., Gagnon, M.P., Griffiths, F., Nicolau, B., O’Cathain, A., Rousseau, M.C., Vedel, I. (2018). Mixed Methods Appraisal Tool (MMAT), version 2018. Registration of Copyright (#1148552), Canadian Intellectual Property Office, Industry Canada.], now reference #25.  

- Reference previously listed as #27 is still relevant and remains correctly cited in the Introduction. Now, it is the reference #32 (Knapp, M., Thorgrimsen, L., Patel, A., Spector, A., Hallam, A., Woods, B., Orrel, M. (2006). Cognitive stimulation therapy for people with dementia: cost-effectiveness analysis. British Journal of Psychiatry, 118, 574-580).  

- Reference previously listed as #31 was incorrectly labeled as Orrell et al., 2017; the correct citation is Leung et al., 2017 (Leung, P., Yates, L., Orgeta, V., Hamidi, F., Orrell, M. (2017). The experiences of people with dementia and their carers participating in individual cognitive stimulation therapy. International Journal of Geriatric Psychiatry, 32(12), e34–e42.), now reference #35. 

- References previously listed as #59 and #60 [now, #65 (Stern, Y. (2002) What is cognitive reserve? Theory and research application of the reserve concept. Journal of the International Neuropsychological Society, 8(3), 448-460.) and #66 (Stern, Y. (2006) Cognitive reserve and Alzheimer disease. Alzheimer Dis Assoc Disord 20(2), 112-117.)] are now explicitly cited in the current version of the manuscript. 

- Reference previously listed as #70 is accurate and remains cited (in the current version, it is reference #83); however, reference #71 was incorrectly included and has been removed. 

Additionally, we have removed the duplicated references previously listed under both #57 and #58. We are confident that these changes have improved the clarity and accuracy of the reference section. Thank you again for your valuable insights. 

COMMENT 2: Line#252-254: I would change the sentence “From the 6 included studies, factors that may influence the gain resulting from cognitive stimulation intervention were extracted a posteriori then assessed in demographic, cognitive, emotional, social factors and quality of life.” to “From the six included studies, factors that may influence the gain resulting from cognitive stimulation intervention were extracted a posteriori and then assessed in terms of demographic, cognitive, emotional, social factors, and quality of life.” 

RESPONSE 2: Dear Reviewer, thank you for your suggestion regarding the sentence in lines 252–254 in the previous version of the manuscript. We agree that your proposed wording improves the clarity and flow of the sentence. We have adopted your revision and updated the manuscript accordingly. (please, see p.13, R 327-329: “From the six included studies, factors potentially influencing the benefits gained from cognitive stimulation intervention were extracted a posteriori then assessed in terms of demographic, cognitive, emotional, social factors and quality of life”) Thank you once again for your careful reading and helpful feedback. 

COMMENT 3: Line#272: the experimental “groupp" with low baseline scores in cognitive performance (and fewer” Change “groupp” to “group”. 

RESPONSE 3: Dear Reviewer, we thank the reviewer for pointing out this typographical error and for the careful attention to detail. However, in response to a suggestion from another reviewer, we have substantially revised the Results section to improve clarity. As a result, this specific sentence has been reformulated and the term “group” no longer appears in that part of the manuscript. 

COMMENT 4: Line#334: According to Stern (2009) there is a point after which “reserve” can no longer with. I would change it to “According to Stern (2009) there is a point after which “cognitive-reserve” can no longer with. 

RESPONSE 4: Dear Reviewer, thank you for your helpful suggestion. We have revised the sentence (previously in line 334) to read: “According to Stern (2009), there is a point after which ‘cognitive reserve’ can no longer withstand the progression of neuropathology.” (see p. 14, R 397). This change improves clarity and accurately reflects the concept discussed. 

COMMENT 5: Line#344: Please correct the typo. “Another finding “in” that low baseline level of cognitive functions predicted improvements in cognition”. “Another finding “is/was” that low baseline level of cognitive functions predicted improvements in cognition”.  

RESPONSE 5: Dear Reviewer, thank you for pointing out this typo. The sentence has been corrected to: “Another finding was that low baseline levels of cognitive functioning predicted improvements in cognition.” (please, see p.16, R 408). We appreciate your attention to detail. 

COMMENT 6: Line#364-366: Please correct the typo. “It is also known that “conducing” the intervention in groups can encourage social interaction (Buschert et al., 2011; Olazaran et al., 2004) and increases the cognitive gain.” “It is also known that “conducting” the intervention in groups can encourage social interaction (Buschert et al., 2011; Olazaran et al., 2004) and increases the cognitive gain.” 

RESPONSE 6: Dear Reviewer, thank you for highlighting this typo. We have corrected “conducing” to “conducting” in the sentence, which now reads: “It is also known that conducting the intervention in groups can encourage social interaction (Buschert et al., 2011; Olazaran et al., 2004) and increases the cognitive gain.” (please, see p.16, R 429). We appreciate your careful reading and constructive feedback. 

COMMENT 7: Line#375-376: For clarity, please revise this sentence “Analogue results if it is considered another form of relationship, that is closer relationship or marital relationship.” to "Analogous results are observed when considering other types of relationships, such as a closer relationship or a marital relationship." 

RESPONSE 7: Dear Reviewer, thank you for your helpful suggestion. We agree that your proposed revision improves the clarity of the sentence. We have amended the text accordingly. The revised sentence now reads: “Analogous results are observed when considering other types of relationships, such as a closer relationship or a marital relationship.” (please, see p.16, R 439-440). We appreciate your attention to clarity and precision. 

COMMENT 8: The authors may consider deleting this repetition. Line#358 : BPSD (behavioral and psychiatric symptoms of dementia)  Line#420 behavioral and psychiatric symptoms of dementia (BPSD) 

RESPONSE 8: Dear Reviewer, thank you for pointing out the redundancy in the definition of the acronym BPSD. We agree with your suggestion and have removed the repeated explanation previously at Line #420, keeping only the first occurrence where the acronym is defined. In the current version of the manuscript, you can see this at p. 16 R 422 and p.18 R515. We appreciate your careful reading and helpful feedback. 

COMMENT 9: Conclusion: Line#427 I have a concern regarding the generalization of findings related to the differential response to cognitive stimulation intervention based on demographic characteristics. Since the observation that women showed greater improvement and maintenance in memory performance over time compared to men is based on limited evidence (only one or two studies), I have a comment regarding the statement, 'Female older people with low formal education level, low baseline level of cognitive functions and lower depressive symptoms have greater improvements from cognitive stimulation intervention,' I suggest rephrasing it to: 'Collectively, these RCT observations suggest that older female gender, individuals with low formal education level, low baseline level of cognitive functions, and lower depressive symptoms may have greater improvements from cognitive stimulation intervention.' The could provide limited scope of the evidence and avoids overgeneralization. 

RESPONSE 9: Dear Reviewer, thank you very much for your thoughtful comment. We agree with your observation regarding the limited number of studies supporting the differential response to cognitive stimulation interventions based on gender. In response to your suggestion, we have revised the phrasing of the statement to avoid overgeneralization and to better reflect the scope of the available evidence. Specifically, we have rephrased the sentence as follows: “Collectively, these RCT observations suggest that older female gender, individuals with low formal education level, low baseline level of cognitive functions, and lower depressive symptoms may have greater improvements from cognitive stimulation interventions.” This revised version has been included in the updated manuscript to ensure a more cautious and accurate interpretation of the findings (please, see p.18 R 544-547). Thank you again for your valuable feedback. 

Reviewer 4 Report

Comments and Suggestions for Authors

See word document

Comments on the Quality of English Language

The manuscript writing style seems somewhat disjointed and does not flow. At times, there are what appear to be fragmented sentences and misplaced sentences that make it somewhat confusing to follow along. Major editorial revisions are recommended. 

Author Response

REVIEWER 4 

Overall Review  

The authors dive into an evaluation of “cognitive stimulation” as a non-pharmacological treatment modality for improving cognitive functioning in older adults. They go on to focus examining the cognitive and non-cognitive factors that contribute to the treatments efficacy. This study seems somewhat confusing regards to their examination of cognitive and non-cognitive factors that impact the benefit of CST for older adults. For example, as it relates to cognition, the authors do not provide great clarity as to how “cognition” is measured to determine which studies are included in their analysis. Cognition is a broad term and their inability to accurate define this factor limits of the # of studies that were used in their analysis and thus limiting the generalizability of their findings.  

We appreciate the Reviewer’s observation regarding the need for greater clarity in how “cognition” was conceptualized and measured in our review. We agree that cognition is a broad construct encompassing a variety of abilities, and that without clear operational definitions, inclusion criteria may appear ambiguous or overly restrictive. To address this, we revised the manuscript to explicitly define what we referred to as cognitive outcomes. Specifically, we distinguished between: 

  • Global cognitive functioning, assessed through standardized cognitive screening tools such as the MMSE, MoCA, or ADAS-Cog, which provide a general index of cognitive status across multiple domains; 
  • And domain-specific cognitive functions, such as memory, attention, executive function, or language, assessed using targeted neuropsychological instruments. 

We clarified that included studies had to evaluate cognitive functioning using either global or domain-specific measures, and had to assess these outcomes pre- and post-intervention. Additionally, eligible studies needed to analyze the relationship between individual characteristics (e.g., age, education, baseline cognition) and intervention outcomes, as the goal of this review was to investigate which person-level factors may moderate the effectiveness of cognitive stimulation. We acknowledge that our focus on studies that examined moderators of response — and not just main group effects — resulted in more stringent eligibility criteria, and possibly a smaller number of included studies. However, we believe this focus adds value by aligning with current trends in dementia care that emphasize individualized and person-centered approaches (Clare et al., 2019; Yates et al., 2016). Below, we now respond point-by-point to each of the Reviewer’s specific comments in the hope of addressing all uncertainties and providing greater clarity on our methodological decisions. 

Introduction  

COMMENT 1: Page 1 line 28: The use of the term “scientific” is extremely broad, its usage seems inappropriate for this context, and does not clearly convey why life expectancy is increasing.  

RESPONSE 1: Dear Reviewer, thank you for your insightful comment. We agree that the term “scientific” was too broad and not sufficiently specific in this context. Accordingly, we have revised the sentence to better reflect the contributing factors to increasing life expectancy. The sentence now reads: “Life expectancy is increasing due to advancements in healthcare, improved living conditions, public health policies, and better disease prevention: according to the World Health Organization (WHO), the global population over the age of 60 is projected to reach 2 billion by 2050.” (please, see p.1, R 28-31). We believe this revision provides a clearer and more accurate explanation. 

COMMENT 2: Page 1 lines 30-32: This sentence seems awkwardly worded and should be broken up into at least two sentences. Also, this sentence could be read as saying that living longer causes age-related diseases as opposed to there being a higher prevalence of diseases in older age. Please consider revising for clarity.  

RESPONSE 2: Dear Reviewer, thank you for your valuable suggestion regarding the clarity and structure of the sentence. We have revised the original sentence to avoid ambiguity and improve readability by splitting it into two sentences. The updated text now reads: “Consequently, the continuous rise in life expectancy has contributed to a growing prevalence of age-related diseases. Among these, dementia represents one of the most significant challenges in terms of diagnosis, treatment, and care (ISTAT, 2018), making it a major concern for current health-care systems.” (please, see p.1, R 31-34) This revision clearly indicates that increased prevalence of age-related diseases is associated with aging, rather than implying that living longer directly causes these diseases. We believe the new wording enhances clarity and maintains an academic yet accessible tone. 

COMMENT 3: Page 1 line 33: Provide citation for WHO Global Action Plan 2017-2025.  

RESPONSE 3: Dear Reviewer, thank you for your comment. We have added the appropriate citation for the WHO Global Action Plan 2017–2025 in the manuscript as follows: (Global action plan on the public health response to dementia 2017–2025. World Health Organization, 2017). This reference is now included both in the text and in the reference list (please, see p.1, R 35-36) and reference #84. 

COMMENT 4: Page 1 lines 38-41: The statement “improvement in cognition and in activities of daily living” appears to be a slight mischaracterization of the findings presented in the citation (Livingston et al. 2024) used to make this statement. cholinesterase inhibitors do not necessarily “improve” cognition as they do not reverse the effects of neurodegenerative pathology but rather slow down the progression of decline seen in cognition and ADL management. The term “improvement” should be changed to better reflect this notion.  

RESPONSE 4: Dear Reviewer, thank you for your valuable observation. We have revised the sentence to more accurately reflect the findings of Livingston et al. (2024). Specifically, we replaced “improvement in cognition and in activities of daily living” with “a slower decline in cognition and in activities of daily living,” which better represents the effect of cholinesterase inhibitors and memantine. The revised sentence now reads: “The available drugs for the treatment of cognitive symptoms in people with dementia (particularly Alzheimer’s disease and Lewy body dementia) are cholinesterase inhibitors (donepezil, rivastigmine, galantamine) and memantine. These are associated with a reduction in symptom severity, and with a slower decline in cognition and in activities of daily living, as well as decreased mortality in individuals with both mild-to-moderate and severe dementia (Livingston et al., 2024).” (please, see p.1-2, R 39-44). We appreciate your careful reading and suggestions to improve the accuracy of our manuscript. 

COMMENT 5: Page 2 lines 43-44: The authors state “monoclonal antibodies drugs can slow down cognitive decline and memory loss in individuals with early-stage Alzheimer’s.” Assuming that they are using the Livingston et al. 2024 article, Livingston and colleagues state that: The approvals were based partly on the “reasonable expectation” that a reduction in amyloid- PET load, or plaques, were likely to predict clinical benefit, although correlations between reduction in plaques and change on clinical ratings scales are very weak.  

Therefore, lines 43-44 seem to mischaracterize the actual effectiveness of their ability to “slow down cognitive decline and memory loss” and should be removed.  

RESPONSE 5: Dear Reviewer, thank you for your insightful comment regarding the characterization of monoclonal antibodies in our manuscript. We have revised the sentence to more accurately reflect the current evidence as reported by Livingston et al. (2024). Specifically, we removed the statement that these drugs can definitively “slow down cognitive decline and memory loss” and replaced it with a more cautious phrasing highlighting the conditional approval, the reduction in amyloid-β plaques, and the limited evidence of meaningful clinical benefit. The revised sentence now reads: “The latest significant advance in disease-modifying therapies for dementia – particularly Alzheimer’s disease – is the conditional approval of donanemab and lecanemab, two monoclonal antibodies that reduce amyloid-β plaque levels. However, evidence supporting a meaningful clinical benefit – such as significantly slowed cognitive decline – remains limited, and the association between amyloid reduction and improved clinical outcomes is weak (Livingston et al., 2024).” (please, see p.2, R44-50). We appreciate your careful review and suggestions which have helped to improve the accuracy of our manuscript. 

COMMENT 6: Page 2 line 78: “Global cognition” needs clarification. From a neuropsychological perspective, the terms listed after global cognition (i.e., memory, language, etc.) are subsumed to be included under the umbrella of global cognition. Provide clarification on the difference between interventions that improve global cognition vs. other types of cognition.  

RESPONSE 6: Dear Reviewer, thank you for your helpful comment regarding the term “global cognition.” To address both your observation and a similar suggestion from another reviewer, we have revised and clarified this section of the manuscript. Specifically, we now use the term “global cognition” to refer to cognitive functioning as assessed by general cognitive screening tools such as the MMSE, ADAS-Cog, or MoCA. These instruments provide a composite score reflecting overall cognitive performance rather than domain-specific abilities. In contrast, we use the term “specific cognitive domains” to refer to studies that evaluated the effects of cognitive stimulation interventions using domain-specific neuropsychological tests focused on particular functions such as memory, language, attention, or executive functioning. This distinction has been made explicit in the revised version of the manuscript to avoid ambiguity and to distinguish between studies assessing general cognitive outcomes and those evaluating domain-specific cognitive functions. The following clarifying phrase has been added to the text: “...intended here as overall cognitive functioning measured through standardized cognitive screening tools that provide a general overview of cognitive abilities without focusing on specific domains.” (please, see p.2, R 67-69). We appreciate your observation, which helped improve the clarity of this important aspect of the review. 

COMMENT 7: Page 2 line 89: Please provide clarification as to what the authors mean when describing “non-cognitive aspects of people with dementia.” When discussing cognitive vs. non-cognitive factors that influence the effectiveness of cognitive stimulation, all of the factors listed by the authors can be assumed to in fact involve “cognition.”  

RESPONSE 7: Thank you for your insightful comment. We acknowledge that the term non-cognitive aspects may appear ambiguous and agree that further clarification was needed. In the revised version of the manuscript, we have clarified that non-cognitive factors refer to individual characteristics not directly related to cognitive performance, such as emotional status (e.g., depressive symptoms), psychosocial aspects (e.g., social engagement, marital status), and demographic variables (e.g., age, sex, education). These are factors that may influence how individuals respond to cognitive stimulation, yet are not typically assessed through neuropsychological or cognitive tests. To improve clarity, we have revised the sentence as follows: “One possible explanation for this variability may lie in the individual characteristics of people with dementia, including both cognitive factors (e.g., baseline cognitive performance) and non-cognitive ones (e.g., emotional status, social engagement, demographic variables), which may influence the outcomes of such interventions.” (please, see p.2, R82-85). This adjustment is intended to better delineate the distinction we make between cognitive and non-cognitive predictors within the context of the review. 

COMMENT 8: Page 3 line 95: Poorly worded and fragmented sentence.  

RESPONSE 8: Thank you for your valuable observation. We agree that the original sentence was poorly worded and lacked clarity. As suggested, we have revised it to improve both grammar and readability. The sentence now reads (please, see p.2, R 88-89): “In recent years, research has increasingly focused on the personalization of cognitive interventions.”. This revised version more accurately conveys the intended meaning and improves the flow of the paragraph. 

COMMENT 9: Page 3 lines 93-105: This paragraph is somewhat confusing. It would be helpful for the authors to provide examples of "personalization of tailored activities" to help the reader understand their purpose of differentiating cognitive and non-cognitive factors.  

RESPONSE 9: Thank you for the observation. We would like to clarify that the aim of this paragraph was not to differentiate between cognitive and non-cognitive factors, but rather to emphasize the increasing focus on the personalization of cognitive interventions in dementia care. Specifically, we aimed to highlight that a better understanding of both cognitive and non-cognitive characteristics may support the development of tailored interventions, which are more effective and meaningful for individuals with dementia. To improve clarity, we have revised the paragraph and included explicit examples of how personalization can be implemented based on individual characteristics (e.g., tailoring activities to one's cognitive abilities, emotional state, or personal preferences). This revision aims to ensure the reader clearly understands the relevance of individual factors—whether cognitive or not—in guiding more effective, person-centered interventions. Please, see p.3, R 93-110: “Evidence from dementia-specific RCTs and reviews supports the efficacy of personally tailored activities, which are adapted based on individual interests, functional abilities, cognitive functioning, or emotional needs. For example, tailoring may involve adapting tasks to one's level of cognitive functioning, personal interests, or emotional needs. Personalized activities are associated with greater engagement and adherence, as they make the experience more meaningful and empowering for people with dementia, promoting participation and improved outcomes (Lu et al., 2021; Warmoth et al., 2022; Booth et al., 2023). A Cochrane review of community-dwelling individuals with dementia found moderate evidence that such personalized interventions reduce challenging behavior and slightly improve quality of life (Parke & Hunter, 2023; Gitlin et al., 2020). Additionally, a case report by Orgeta et al. (2015) demonstrated that a culturally and psychologically personalized cognitive stimulation program enhanced engagement, symptom acceptance, and mild cognitive benefit. Furthermore, the SADEM trial in adults with mild dementia showed that a long-term (12-month) multicomponent cognitive intervention produced improvements in cognition and daily living activities (Hsieh et al., 2024). These findings underscore the importance of personalization, even in traditional non-pharmacological interventions, to enhance relevance and effectiveness for individuals with dementia.” 

Methods  

COMMENT 10: Page 4 line 146: The author’s reference “other medical conditions” as an exclusionary criteria for studies studying patients with dementia. What does “ other medical conditions?” Generally speaking the majority of older adults with dementia are going to have at least 1+ comorbid medical condition. Does that mean that these individuals were excluded? This may limit the number of studies that were included in their analysis.  

RESPONSE 10: We thank the Reviewer for this important observation. To clarify, our exclusion criterion regarding "other medical conditions" was not intended to exclude individuals with common age-related comorbidities, which are indeed prevalent among people with dementia. Rather, we excluded studies involving participants with medical or psychiatric conditions that could confound or obscure the diagnosis of dementia as the primary condition—for example, major psychiatric disorders, stroke, or traumatic brain injury. This decision is consistent with standard diagnostic criteria for dementia, as outlined in the DSM-5 and ICD-10, which require that cognitive decline not be better explained by other neurological, psychiatric, or systemic conditions (APA, 2013; WHO, 1992). Our objective was to ensure that included studies focused specifically on individuals with a confirmed diagnosis of dementia, in accordance with these diagnostic standards. To enhance clarity, we have revised the manuscript to explicitly state this rationale in the Methods section (please, see p. 5, R196-199: “This last decision was grounded in standard diagnostic criteria for dementia. DSM-5 and ICD-10 require that cognitive decline not be better explained by other neurological, psychiatric or systemic medical disorders (APA, 2013; WHO, 1992)”). 

COMMENT 11: Page 4 lines 150-151: “Studies that did not measure cognitive and non-cognitive out-150 comes or that measured improvement in single cognitive functions.” Authors do not attempt to clarify the difference between cognitive functioning included vs. a single cognitive domain. Again based on their exclusion criteria this may be a factor that limits the number studies that were analyzed.

RESPONSE 11: We appreciate the Reviewer’s comment and the opportunity to clarify this point. Our intention in excluding studies that focused solely on a single cognitive function was grounded in the specific objectives of our review. Rather than summarizing general effects of cognitive stimulation, our aim was to examine how individual cognitive and non-cognitive characteristics influence the effectiveness of such interventions. To do so meaningfully, we specifically included studies that evaluated the effects of cognitive stimulation on multiple cognitive domains rather than on a single cognitive function because dementia typically affects a range of cognitive abilities simultaneously. Measuring improvement across multiple domains allows for a more comprehensive and ecologically valid assessment of the intervention’s effectiveness. Focusing solely on one cognitive domain may provide a limited view of the intervention’s impact and fail to capture broader cognitive changes that are relevant to daily functioning and quality of life. Additionally, cognitive stimulation interventions are often designed to target general cognitive functioning or multiple domains (e.g., memory, attention, executive functions), and thus outcome assessments reflecting this multidimensionality provide a better evaluation of their true clinical benefit. Therefore, studies that measured only improvement in a single domain (e.g., attention only), without considering broader effects or without analyzing how participant characteristics influenced those outcomes, were excluded. This methodological choice reflects our focus on person-centered care approaches, which emphasize understanding for whom and under which conditions interventions are most beneficial (Clare et al., 2019; Yates et al., 2016). To address this more explicitly, we revised the exclusion criteria in the manuscript as follows: “Studies that did not include cognitive and non-cognitive outcomes, that focused solely on the improvement of a single cognitive function, or that did not explore the relationship between individual characteristics and intervention outcomes (e.g., studies reporting only overall group-level effects, without analysis of influencing factors)”, please see p.5 R 223-226.  We also clarified the rationale in the Inclusion Criteria subsection to better communicate this distinction and to acknowledge that this approach may have narrowed the pool of eligible studies, but was necessary to align with the specific aims of our review (please, see p. 4, R 170-179: “We specifically included studies that evaluated the effects of cognitive stimulation on multiple cognitive domains rather than on a single cognitive function because dementia typically affects a range of cognitive abilities simultaneously. Measuring improvement across multiple domains allows for a more comprehensive and ecologically valid assessment of the intervention’s effectiveness. This approach aligns with existing literature emphasizing the importance of multi-domain cognitive assessments in dementia research to detect meaningful and generalizable changes (Clare & Woods, 2004; Yates et al., 2016). Therefore, our inclusion criteria aimed to capture studies that evaluate cognitive outcomes more comprehensively to inform the development of personalized and effective cognitive stimulation interventions.”). 

Results:  

Discussion 

Reviewer 5 Report

Comments and Suggestions for Authors

The manuscript systematically examined the potentially influence factors that may affect the effectiveness of cognitive stimulation on the cognitive and non-cognitive performance for older patients with dementia.  This is a very well-written and interesting paper.  However, The title in this paper seem to amplify the research topic. 1) personalized cognitive interventions was not the research topic, authors did not compare the effectiveness of different CS methods; 2) this study only includes patients with dementia

I have some minor suggestions for authors.

  1.  the title “...in people with cognitive impairments” suggest replace with "...in people with dementia"
  2. to suggest systematically review effect of CS on domain-specific cognitive performance for older people with dementia
  3. About the exclusion criteria b) “isolated cognitive stimulation interventions on a single cognitive function”, I suggest authors systematically review relative literature that add evidence for this research topic.
  4. Table 2 and Table 3, I suggest add a column of conclusions, overall quality in table 2, and topic-related conclusion in Table 3.   

Author Response

REVIEWER 5 

The manuscript systematically examined the potentially influence factors that may affect the effectiveness of cognitive stimulation on the cognitive and non-cognitive performance for older patients with dementia.  This is a very well-written and interesting paper.  However, The title in this paper seem to amplify the research topic. 1) personalized cognitive interventions was not the research topic, authors did not compare the effectiveness of different CS methods; 2) this study only includes patients with dementia 

I have some minor suggestions for authors. 

COMMENT 1:  the title “...in people with cognitive impairments” suggest replace with "...in people with dementia" 

RESPONSE 1: We thank the Reviewer for the kind and constructive feedback. In response to the Reviewer’s suggestion regarding the title, we agree that the original formulation ("...in people with cognitive impairments") was too broad, considering that our review exclusively focuses on individuals with dementia. Furthermore, we acknowledge the important observation that our paper does not compare different types of cognitive stimulation methods, nor does it directly evaluate personalized interventions. Rather, it aims to identify cognitive and non-cognitive variables that may influence individual response to cognitive stimulation, which can inform future personalized approaches.  Accordingly, we have revised the title as follows: “Cognitive and non-cognitive predictors of response to cognitive stimulation in dementia: a systematic review toward personalization”. This new formulation more accurately reflects the scope of the study. It emphasizes the key focus on identifying influencing factors (cognitive and non-cognitive), clearly defines the population of interest (people with dementia), and places the notion of personalization as a future-oriented implication — aligning with the Reviewer’s helpful suggestion. We hope the revised title addresses the concern and provides a clearer representation of the manuscript’s contribution. 

COMMENT 2: to suggest systematically review effect of CS on domain-specific cognitive performance for older people with dementia 

RESPONSE 2: Thank you for the insightful comment. We specifically included studies that evaluated the effects of cognitive stimulation on multiple cognitive domains rather than on a single cognitive function because dementia typically affects a range of cognitive abilities simultaneously. Measuring improvement across multiple domains allows for a more comprehensive and ecologically valid assessment of the intervention’s effectiveness. Focusing solely on one cognitive domain may provide a limited view of the intervention’s impact and fail to capture broader cognitive changes that are relevant to daily functioning and quality of life. Additionally, cognitive stimulation interventions are often designed to target general cognitive functioning or multiple domains (e.g., memory, attention, executive functions), and thus outcome assessments reflecting this multidimensionality provide a better evaluation of their true clinical benefit. This approach aligns with existing literature emphasizing the importance of multi-domain cognitive assessments in dementia research to detect meaningful and generalizable changes (Clare & Woods, 2004; Spector et al., 2010; Yates et al., 2016). Therefore, our inclusion criteria aimed to capture studies that evaluate cognitive outcomes more comprehensively to inform the development of personalized and effective cognitive stimulation interventions. Thank you for raising this point. We have since expanded criterion d in the inclusion criteria section to offer a clearer and more comprehensive explanation, please see  p. 4, R 170-179 of the recent version of our manuscript: “We specifically included studies that evaluated the effects of cognitive stimulation on multiple cognitive domains rather than on a single cognitive function because dementia typically affects a range of cognitive abilities simultaneously. Measuring improvement across multiple domains allows for a more comprehensive and ecologically valid assessment of the intervention’s effectiveness. This approach aligns with existing literature emphasizing the importance of multi-domain cognitive assessments in dementia research to detect meaningful and generalizable changes (Clare & Woods, 2004; Yates et al., 2016). Therefore, our inclusion criteria aimed to capture studies that evaluate cognitive outcomes more comprehensively to inform the development of personalized and effective cognitive stimulation interventions.“). 

COMMENT 3: About the exclusion criteria b) “isolated cognitive stimulation interventions on a single cognitive function”, I suggest authors systematically review relative literature that add evidence for this research topic. 

RESPONSE 3: Thank you for your valuable suggestion. We chose to exclude studies focusing on isolated cognitive stimulation interventions targeting a single cognitive function because the existing literature, including seminal works such as Clare and Woods (2004), clearly differentiates cognitive stimulation from other approaches like cognitive training or cognitive rehabilitation. Cognitive stimulation is defined as a multi-domain intervention aiming to engage several cognitive functions simultaneously, which is considered more effective in improving global cognitive functioning in people with dementia. Several systematic reviews and meta-analyses support the greater effectiveness of multi-domain cognitive stimulation compared to interventions focused on a single cognitive domain. For example, Woods et al. (2012), in their Cochrane review, emphasized that cognitive stimulation interventions typically target multiple cognitive domains, producing significant improvements in global cognition, quality of life, and social functioning, whereas single-domain cognitive training often results in domain-specific gains that do not generalize well to overall cognition or daily functioning. Similarly, Bahar-Fuchs et al. (2019) highlighted that multi-domain interventions better reflect the complex, multifaceted cognitive decline seen in dementia and can therefore induce broader cognitive benefits. They suggest that focusing on a single cognitive domain may overlook important aspects of cognitive decline and limit transfer effects to other cognitive abilities and real-life functioning. Additionally, the person-centered and holistic nature of cognitive stimulation, which targets engagement, social interaction, and multiple cognitive processes simultaneously, is a key factor distinguishing it from isolated cognitive training or rehabilitation (Clare & Woods, 2004). This multi-domain approach aligns with recommendations by the National Institute for Health and Care Excellence (NICE, 2018), which supports interventions addressing multiple cognitive and psychosocial domains to maximize therapeutic benefits in dementia care. For these reasons, and to maintain conceptual clarity and clinical relevance, our review focused exclusively on studies evaluating multi-domain cognitive stimulation interventions. We have clarified this rationale further in the revised manuscript, please see p.5, R 206-217: “The decision to exclude studies focusing on cognitive stimulation targeting a single cognitive domain is supported by existing literature emphasizing the superior efficacy of multi-domain cognitive stimulation interventions. Clare and Woods (2004) clarify that cognitive stimulation is characterized by engaging multiple cognitive domains, distinguishing it from domain-specific cognitive training or rehabilitation. Systematic reviews, including those by Woods et al. (2012) and Bahar-Fuchs et al. (2019), provide evidence that multi-domain cognitive stimulation leads to broader improvements in cognitive functioning and daily living activities compared to interventions focusing on a single cognitive domain. Furthermore, clinical guidelines such as those from NICE (2018) recommend multi-domain approaches as standard practice for cognitive interventions in dementia care, reinforcing the rationale behind our exclusion criteria." 

COMMENT 4: Table 2 and Table 3, I suggest add a column of conclusions, overall quality in table 2, and topic-related conclusion in Table 3.  

RESPONSE 4: Thank you for this thoughtful and constructive suggestion. Regarding Table 2, which presents the risk of bias assessment using the Mixed Methods Appraisal Tool (MMAT 2018), we have decided not to include a column for an overall quality score, in accordance with the MMAT guidance. As clearly stated in the MMAT protocol: “It is discouraged to calculate an overall score from the ratings of each criterion. Instead, it is advised to provide a more detailed presentation of the ratings of each criterion to better inform the quality of the included studies.” (Hong et al., 2018, Mixed Methods Appraisal Tool (MMAT), version 2018 user guide). Following these recommendations, we opted to provide an item-by-item rating in Table 2, as this offers a more nuanced and transparent view of each study’s methodological quality. We believe this approach supports more accurate interpretation and avoids the risk of misleading simplification that could result from aggregated scoring. Of course, we remain open to revising the format if the reviewer has a specific alternative presentation in mind, but we preferred to adhere to established methodological standards. Regarding Table 3, we would like to clarify that this table was specifically designed to present the methodological and structural characteristics of the included studies—namely: authorship, country, study design, total sample, intervention/control group, frequency and duration of the intervention, content of the intervention, pre- and post-assessment, and main results. As for the suggestion to include a "topic-related conclusion" column, if we correctly interpreted the reviewer’s intent, we would like to highlight that this information is systematically presented in Table 4, which synthesizes the cognitive and non-cognitive individual factors found to be associated with greater benefit from cognitive stimulation. Given that our review focuses on identifying these moderators of intervention outcomes, we chose to dedicate a separate table to this thematic synthesis rather than integrating such content into Table 3, in order to avoid redundancy and maintain clarity.

Round 2

Reviewer 2 Report

Comments and Suggestions for Authors

The authors have adequately addressed all the concerns previously raised, enhancing the clarity, depth, and rigor of the manuscript. It is now well-prepared for publication.

Reviewer 4 Report

Comments and Suggestions for Authors

Thank you for the opportunity to review this manuscript. The authors have addressed my concerns adequately.